



# Analysis of the long-range transport of the volcanic plume from the 2021 Tajogaite/Cumbre Vieja eruption to Europe using TROPOMI and ground-based measurements

Pascal Hedelt[1,*], Jens Reichardt[2,*], Felix Lauermann[2], Benjamin Weiß[1], Nicolas Theys[3], Alberto Redondas[4], Africa Barreto[4], Omaira Garcia[4], and Diego Loyola[1]

[1]Deutsches Zentrum für Luft- und Raumfahrt, Remote Sensing Institute (DLR-IMF), Germany
[2]Deutscher Wetterdienst (DWD), Meteorologisches Observatorium Lindenberg, Germany
[3]Royal Belgian Institute for Space Aeronomy (BIRA-IASB), Brussels, Belgium
[4]Izaña Atmospheric Research Center, AEMET- State Meteorological Agency, Spain
[*]These authors contributed equally to this work.

**Correspondence:** Pascal Hedelt (pascal.hedelt@dlr.de), Jens Reichardt (Jens.Reichardt@dwd.de)

**Abstract.** The eruptions of the Tajogaite volcano on the western flank of the Cumbre Vieja ridge on the island of La Palma between September and December 2021 released large amounts of ash and $SO_2$. Transport and dispersion of the volcanic emissions were monitored by ground-based stations and satellite instruments alike. In particular, the spectrometric fluorescence and Raman lidar RAMSES at the Lindenberg Meteorological Observatory measured the plume of the strongest Tajogaite eruption of 22-23 September 2021 over northeastern Germany four days later. This study provides an analysis of $SO_2$ vertical column density (VCD) and layer height (LH) measurements of the volcanic plume obtained with Sentinel-5 Precursor/TROPOMI, which are compared to the observations at several stations across the Canary Islands. Furthermore, a new modeling approach based on TROPOMI $SO_2$ VCD measurements and the HYSPLIT trajectory and dispersion model was developed which confirmed the link between Tajogaite eruptions and Lindenberg measurements. Modeled mean emission height at the volcanic vent is in excellent agreement with co-located TROPOMI $SO_2$ LH and local lidar ash height measurements. Finally, a comprehensive discussion of the RAMSES measurements is presented. A new retrieval approach has been developed to estimate the microphysical properties of the volcanic aerosol. For the first time, an optical particle model is utilized that assumes an irregular, non-spheroidal shape of the aerosol particles. According to the analysis, the volcanic aerosol consisted solely of fine-mode inorganic, solid and irregularly shaped particles - the presence of large aerosol particles or wildfire aerosols could be excluded. The particles likely had an isometric to slightly plate-like shape with an effective half of particle maximum dimension around 0.1 µm and a refractive index of about 1.51. Moreover, mass column values between 70 and 110 $\mathrm{mg\,m^{-2}}$, mean mass concentrations of 45-70 $\mathrm{\mu g\,m^{-3}}$, and mean mass conversion factors between 0.21 and 0.33 $\mathrm{g\,m^{-2}}$ at 355 nm were retrieved. Possibly RAMSES observed, at least in part, volcanic secondary sulfate aerosol which was produced by gas-phase homogeneous reactions during the transport of the air masses from La Palma to Lindenberg.



## 1 Introduction

Volcanic eruptions emit large amounts of particulate matter and trace gases into the atmosphere, which can have a major impact on human health, society, and nature. While the main concern related to volcanic ash plumes is air traffic safety, of the various trace gases emitted such as sulphur species, water vapor, carbon dioxide, and halogens, sulphur dioxide ($SO_2$) has received particular attention due to its subsequent conversion to aerosols (see e.g., Rix et al., 2012) and its potentially strong effect on global climate (Robock, 2000). $SO_2$ is also the volcanic gas that is most easily detected using ultraviolet (UV) and thermal infrared remote-sensing techniques, and has therefore been used to monitor volcanoes worldwide for many decades (see e.g., Rix et al., 2009; Carn et al., 2016, 2021; Prata and Lynch, 2019; Coppola et al., 2020).

Global networks of ground-based stations enable the monitoring of air quality, atmospheric trace gases and atmospheric constituents, including aerosols, clouds, ozone ($O_3$), nitrogen dioxide ($NO_2$), formaldehyde (HCHO) and $SO_2$, with high temporal resolution and accuracy. However, these measurement sites are mainly located in the region of interest, i.e. in or near urban areas or at volcanoes that are potentially hazardous to nearby settlements to allow long-term monitoring at close range. Unfortunately, only 45 % of known active volcanoes are monitored operationally from the ground, as most volcanoes are inaccessible (see e.g., Sparks et al., 2012; Widiwijayanti et al., 2024). Satellite sensors do not have this limitation, they can make observations on a global scale and are therefore ideal for detecting eruptions and tracking the movement of volcanic clouds, albeit with a limited parameter set and less accuracy than the ground stations. Complementary measurements with ground-based and satellite-based instruments would therefore be ideal for a more comprehensive picture.

Not without earlier warnings (Torres-González et al., 2020) and after a swarm of earthquakes a week before to the eruption, several vents opened on 19 September 2021 at 14:10 UTC at the Cumbre Vieja volcanic ridge on La Palma in the Canary Islands, Spain (IGN). Different eruptive styles were observed, ranging from strombolian to effusive with occasional strong ash-rich explosions, lava effusion, ash and gas jets (Carracedo et al., 2022; Romero et al., 2022). According to reports from the Toulouse Volcanic Ash Advisory Center (VAAC), the ash plumes rose up to 5.5 km in height. The eruption, which was subsequently referred to as the 'Tajogaite' eruption, ended on 13 December 2021 after 85 days and was finally declared over on 25 December 2021 by the Steering Committee of the Special Plan for Civil Protection and Attention to Emergencies due to Volcanic Risk (PEVOLCA). The eruption had a strong impact on public health and the economy: about 7,000 people had to be evacuated, and the lava flows covered an area of about 12.2 $km^2$, destroying settlements, towns and banana plantations (Carracedo et al., 2022; PEVOLCA). According to calculations by the Canary Islands government, the economic damage was estimated at over 800 million euros.

Volcanic discharge of lava and $SO_2$ was most intense in the first weeks of the eruption. Measurements with the TROPO-spheric Monitoring Instrument (TROPOMI) aboard the European Space Agency (ESA) Sentinel-5 Precursor (S5P) satellite showed that daily emission of $SO_2$ was maximum on 23 September 2021 ($125 \text{ kt d}^{-1}$), remained at elevated levels until 7 November, and decreased significantly afterwards (Milford et al., 2023). While the ash cloud was confined to the Canary Islands archipelago (Salgueiro et al., 2023), the $SO_2$ plume eventually covered vast areas (Filonchyk et al., 2022) and extended northeast at least to central Europe and south to Cabo Verde (Gebauer et al., 2023).

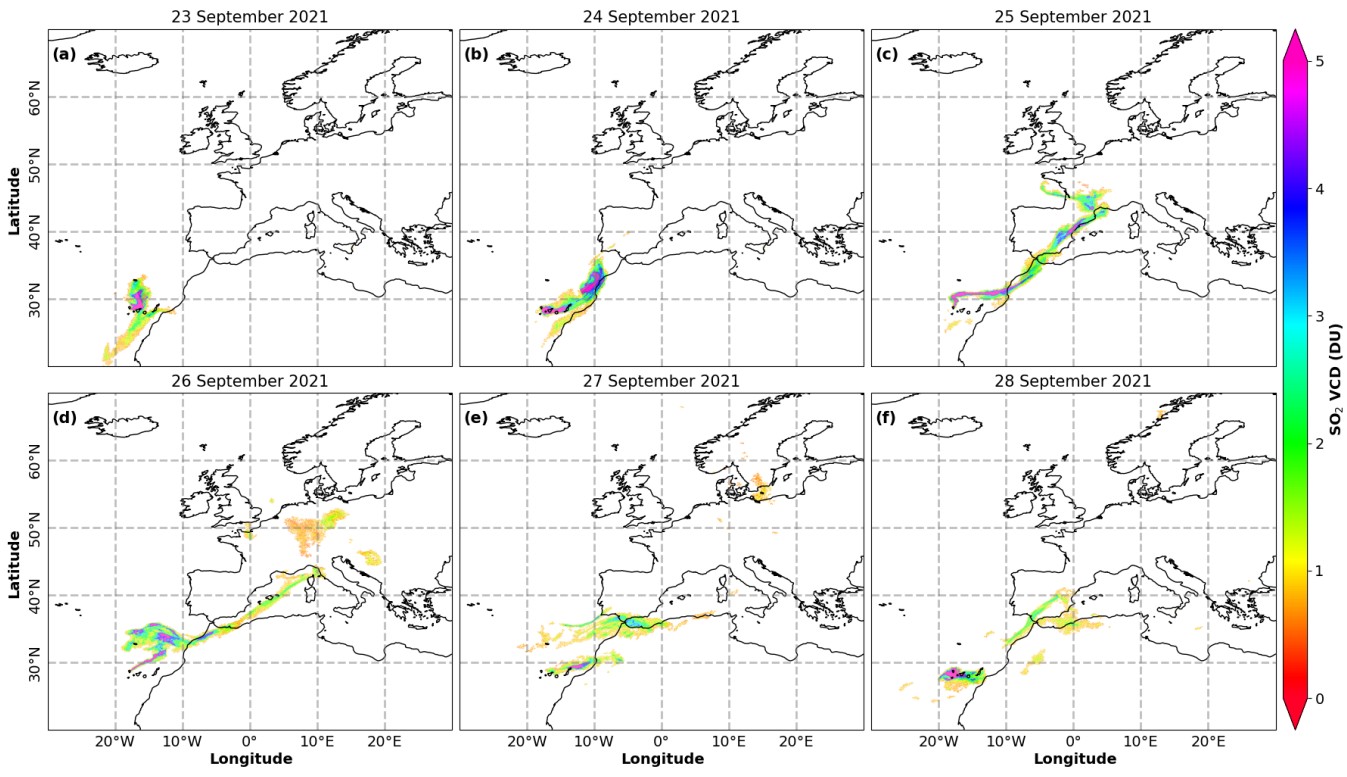

**Figure 1.** Daily TROPOMI SO$_2$ VCD measurements of the volcanic plume from the Tajogaite/Cumbre Vieja eruption during 23-28 September 2021. Only pixels with enhanced SO$_2$ amounts are shown.

The temporal and spatial development of the Tajogaite emissions was monitored over the entire period of volcanic activity
by a considerable number of ground stations and satellites. Scientific interest was so great that additional instrumentation was
even brought to La Palma. Several studies have already been published. The investigations address the impact of the eruptions
on air quality (Filonchyk et al., 2022; Sicard et al., 2022; Milford et al., 2023), and the microphysical and radiative properties
of the volcanic cloud (Bedoya-Velásquez et al., 2022; Córdoba-Jabonero et al., 2023; García et al., 2023; Gebauer et al., 2023;
Salgueiro et al., 2023). However, probably in part due to instrument commissioning time, not all of them study the intense
September phase of volcanic activity.

In these first days of the eruption, SO$_2$ vertical column density (VCD) as observed by TROPOMI reached values of more than
10 DU in the vicinity of the Canarian Islands. The volcanic plume was transported mostly in northeastern direction towards
Europe where it slowly dispersed over Germany and southern Scandinavia and eventually disappeared after 27 September
2021. Figure 1 shows the daily SO$_2$ VCD of the volcanic cloud measured by TROPOMI. A strong eruptive event on 22–



23 September injected large amounts of $SO_2$ into the atmosphere (Fig. 1a) which were transported over southern Spain and France (Fig. 1b–c), where the plume split. The sections of it that were emitted first were transported in a northeasterly direction towards Germany, while the younger sections of the plume moved over Italy and Spain. (Fig. 1d-f). Previous studies of this event were performed based on data from passive and active instruments operating at ground stations close to Tajogaite on the Canary Islands (Bedoya-Velásquez et al., 2022; García et al., 2023), in Spain (Salgueiro et al., 2023) and southern France

(Bedoya-Velásquez et al., 2022). Unfortunately, only basic ceilometers (some with depolarization measurement capability) were available to all research groups, so that lidar-based analyses were only possible to a limited extent.

This publication is aimed at furthering research into the early northeastern Tajogaite plume. It passed over Germany on 26-27 September 2021 (see Fig. 1d) where it was measured by the instruments of the Lindenberg Meteorological Observatory, most notably by the Raman lidar for atmospheric moisture sensing (RAMSES), the high-performance spectrometric fluorescence

and Raman lidar of the German Meteorological Service (Reichardt et al., 2012), under excellent atmospheric conditions. The paper focuses on this unique measurement case. A trajectory and dispersion model is used to establish the Tajogaite eruption on 23 September 2021 as the source of the aerosol which was subsequently observed over Lindenberg after its long-range transport. The RAMSES measurements of the volcanic aerosol are discussed in detail, and its microphysical properties are estimated using a new retrieval approach. To provide some background, $SO_2$ amounts and volcanic plume heights measured

by ground-based instruments and TROPOMI are compared over the entire eruptive period from September to December 2021 first. The paper is organized as follows. Section 2 describes the datasets and instruments used in this paper, Sect. 3 gives an overview about the methods applied, and in Sect. 4 the results are presented and discussed.

## 2 Datasets and instruments

### 2.1 Satellite data from Sentinel-5P/TROPOMI

The ESA Sentinel-5P satellite was launched in October 2017. Onboard is TROPOMI, a nadir-viewing imaging spectrometer, covering wavelength bands between the UV and the shortwave infrared. It has a swath width of $\sim 2.600$ km on the Earth's surface with a ground-pixel resolution of $5.5 \times 3.5$ km$^2$ (after mid-August 2019) for the UV bands. Based on TROPOMI Earthshine reflectance measurements, near-real time (NRT) retrievals are performed to calculate VCDs for $SO_2$, $O_3$, $NO_2$, HCHO, and other atmospheric trace gases. Due to the broad swath, global daily measurements of the Earth's atmosphere can

be performed, allowing e.g. the monitoring and tracking of volcanic emissions worldwide. The overpass time over the Canary Islands is around 14:00 UTC. In this study, the (reprocessed) TROPOMI Level 2 (L2) $SO_2$ product version 03 is used (ESA S5P, 2022).

$SO_2$ VCDs are operationally retrieved from TROPOMI measurements with the Differential Optical Absorption Spectroscopy (DOAS) method which yields the $SO_2$ slant column density (SCD), i.e. the density along the light path. A so-called air mass

factor is then applied to convert the SCD to VCD. A-priori atmospheric $SO_2$ profiles need to be assumed since the actual altitude of the $SO_2$ layer is challenging to estimate directly from UV measurements. $SO_2$ can be injected into the atmosphere at various injection heights depending on the source (anthropogenic or diffusive, weak, or explosive volcanic activity). Therefore,





the operational $SO_2$ L2 product contains four different VCDs for the following scenarios: an anthropogenic pollution VCD as well as three volcanic VCDs assuming a volcanic $SO_2$ layer located at 1, 7, or 15 km, respectively, thus representing the full range of possible eruptive scenarios. Eventually, the user has to select the most appropriate VCD for a given volcanic eruption. See Theys et al. (2017) for details about the retrieval algorithm.

Only recently new methods have been developed to directly retrieve the actual $SO_2$ layer height (LH) from TROPOMI data. Hedelt et al. (2019) apply a combined Principal Component Analysis (PCA) and Neural Network (NN) approach to determine $SO_2$ LH with an accuracy of $< 2$ km for $SO_2$ VCDs $> 20$ DU. This $SO_2$ LH approach has been validated by Koukouli et al. (2022), and its results are now actively assimilated by the European Center for Medium Weather Forecast (ECMWF) to improve $SO_2$ forecasts, as shown by Inness et al. (2022). In July 2023 it was implemented in the operational TROPOMI $SO_2$ retrieval algorithm. $SO_2$ LHs presented in this paper have been obtained with algorithm version 4.0 (see $SO_2$LH ATBD).

TROPOMI $SO_2$ data contain a detection flag, which flags pixels with enhanced $SO_2$ content above a threshold of 0.35 DU, see Theys et al. (2017). This flag will be used in the following to select pixels of the volcanic plume.

## 2.2 Ground-based instruments on the Canary Islands

A total of five local lidars were used by AEMET, the State Meteorological Agency of Spain, in collaboration with other ACTRIS (Aerosol, Clouds and Trace Gases Research Infrastructure) members in Spain, to monitor the altitude and wind-driven dispersion of the volcanic plume during the entire Tajogaite eruptive episode. Data from the Raman lidar installed at the Astrophysical Observatory Roque de los Muchachos (ORM), a micropulse lidar (type MPL-4B) deployed at Tazacorte (TAZ) and three ceilometers (Lufft CHM-15k at Fuencaliente, FUE; Vaisala CL51 at La Palma Airport, AER; and Vaisala CL61 at El Paso; EP) are used in this paper, see Tab. 1 for details. The retrieval method for AEMET-ACTRIS aerosol LH depends on the type of the instrument; it encompasses a qualitative estimation of the LH, which is based either on the gradient method (Flamant et al., 1997; Sicard et al., 2022) or the continuous Wavelet Covariance Transform (WCT) method (Baars et al., 2008; Bedoya-Velásquez et al., 2022). The AEMET-ACTRIS LH dataset comprises a total of 137 altitudes of the dispersive volcanic plume, a detailed description of the database can be found in Sicard et al. (2022).

Ground-based measurements of $SO_2$ VCDs were performed with a Brewer spectrophotometer during Tajogaite volcanic activity with high temporal resolution. The instrument was operated at the subtropical high-mountain Izaña Observatory (IZO) on Tenerife; it participates in the European Brewer Network (EUBREWNET) and is managed by the Izaña Atmospheric Research Center (IARC), which is part of AEMET.

## 2.3 Ground-based instrument RAMSES in Lindenberg (Germany)

Data from the Lindenberg Meteorological Observatory - Richard Aßmann Observatory (MOL-RAO) in Germany are employed to investigate the volcanic plume traveling over Germany. The observatory is part of the German Meteorological Service (Deutscher Wetterdienst - DWD) and located in the northeastern part of Germany close to Berlin at 52.21 N, 14.12 E. In particular, RAMSES data are used in this study.





**Table 1.** Location of AEMET stations on the Canary Islands

| Name | Abbreviation | Location | Altitude | Distance to Tajogaite |
|---|---|---|---|---|
| Roque de Los Muchachos | ORM | 28.76 N, 17.89 W | 2,423 m | 15 km |
| Tazacorte | TAZ | 28.64 N, 17.93 W | 140 m | 4 km |
| El Paso | EP | 28.65 N, 17.88 W | 700 m | 3 km |
| La Palma Airport | AER | 28.62 N, 17.75 W | 56 m | 20 km |
| Fuencaliente | FUE | 28.49 N, 17.85 W | 630 m | 10 km |
| Izaña | IZO | 28.31 N, 16.50 W | 2,370 m | 144 km |

RAMSES is a spectrometric fluorescence and water Raman lidar. Commissioned in 2005, RAMSES has been continuously expanded over the years and is now one of the most powerful multi-parameter lidar systems in the world (Reichardt et al., 2012). Its unique feature is the operation of three spectrometers, which allows the measurement of water in all its three phases (Reichardt, 2014; Reichardt et al., 2022) and the measurement of the fluorescence spectrum of atmospheric aerosols for their characterization and interaction with clouds (Reichardt et al., 2023). A pulsed (30 Hz) frequency tripled flashlamp-pumped

Continuum Powerlite Precision II 9030 laser serves as the radiation source. A Pellin Broca prism is implemented in the transmitter to ensure that only UV light (355 nm, 500 mJ pulse energy) is emitted, which is a prerequisite for spectral fluorescence measurements. The receiver has two telescopes with diameters of 30 and 79 cm for measurements up to 7 km and above 2 km altitude (depending on parameter), respectively, so the plume of the Cumbre Vieja eruption was within the optimal observation range of RAMSES. Therefore, in addition to humidity and the spectral aerosol properties, the elastic scattering properties

(extinction and backscatter coefficient, lidar and depolarization ratio) are also available in high quality for the analysis of the volcanic event.

## 3    Methods

This section describes all methods used in this work in detail.

### 3.1    Comparisons of ground-based and satellite measurements in the vicinity of the volcanic vent

Ground-based Brewer $SO_2$ VCD datafrom the IZO station on Tenerife are directly compared to TROPOMI $SO_2$ VCD measurements during the satellite overpasses. Only measurements are considered that were taken close to the volcanic vent. Given the fact that the Tajogaite eruptions were not particularly kinetic, low to moderate plume heights of 1 km and 7 km are assumed for TROPOMI $SO_2$ VCDs. In addition, $SO_2$ VCDs for retrieved plume heights are used (i.e. for pixels with high $SO_2$ VCD). The median TROPOMI $SO_2$ VCD of all pixels within a radius of 10 km around the IZO station (Tab. 1) is calculated. It should

be noted, however, that such comparison is a challenging task because of the differing measurement principles involved. For one, the ground-based instruments perform zenith scans from below, while the satellite is nadir-viewing. Therefore both ob-



servations are sensitive to different vertical segments of the volcanic plume. Moreover, due to the different viewing directions, both methods are subject to differential cloud-screening. Depending on the location of a possible meteorological cloud with respect to the plume, it may block the observation path or not. Finally, ground-based instruments have a significantly narrower field-of-view (FOV) compared to satellite instruments, hence the observed horizontal area of the volcanic plume is not the same.

Furthermore, ground-based lidar ash-height measurements on La Palma are contrasted with TROPOMI $SO_2$ LH retrievals. The challenge here is that air parcels containing $SO_2$ are not necessarily co-located with the ash plume even if discharged at the same time because of the differing atmospheric dynamics of particles and gases and the gravitational settling of the heavier ash. The FOV mismatch poses a problem here as well. To minimize the error due to differences in measurement time and FOV, the median TROPOMI $SO_2$ LH over all pixels within a radius of 10 km, or 200 km, around the corresponding lidar is calculated, the maximum allowable time difference is set to 5 h. Note that TROPOMI $SO_2$ LH and SCD data are known to exhibit a low bias in the presence of volcanic ash. Also, the TROPOMI data might be affected by cloud-screening.

## 3.2 Trajectory analysis method

The Hybrid Single-Particle Lagrangian Integrated Trajectory Model (HYSPLIT, Stein et al., 2015) is employed to analyze the transport of the volcanic plume and to investigate the plume height, injection and arrival height and time as well as the particle travel time. Backward trajectory calculations are performed using the GFS0.25 meteorology, which is a daily dataset from the Global Forecast System (GFS) with 0.25° spatial and 6-hour temporal resolution, provided by the National Centers for Environmental Prediction (NCEP, NCEP/GFS0.25, 2015).

The start locations of the trajectories and the release times depended on the type of analysis intended:

1. To determine the layer height of the pixels with enhanced $SO_2$ content observed by TROPOMI over Europe on 26 September 2021, the backward trajectory calculations were started at the TROPOMI observation times. For trajectory start locations, three TROPOMI orbits (orbit numbers 20487 to 20489) were examined. The overpass times were at 10:45 UTC and 12:25 UTC over continental Europe and 14:00 UTC over La Palma, respectively. A total of 32,005 pixels were selected as starting locations for the trajectories, which corresponds to every pixel on that day in the region of interest meeting the imposed requirements of $SO_2$ VCD being $> 0.1$ DU and flagged as enhanced $SO_2$. In order to account for the spatial extent of the pixel and related variations in the meteorological field, eight additional starting locations with latitudinal and longitudinal offsets of $\pm 0.01°$ around the pixel center were created. The trajectories were initialized with starting altitudes between 0.25 and 8.75 km in 50-m increments. The maximum simulation runtime was 7 days.

Trajectories were assigned to the Tajogaite eruption if they passed the volcanic vent within a 100 km radius. Trajectories traveling more than one day were further filtered if they passed at least one TROPOMI pixel with enhanced $SO_2$ VCD on each intermediate day between 23 and 26 September within a range of 50 km (cf. Fig. 1) and within a time interval of $\pm 2$ hours around the measurement time of the pixel. Trajectories were discarded if the altitude at the point of their



closest approach to the volcano was not within 2 to 6.5 km. This height range is roughly oriented to the results of the ground-based measurements described in Sect. 3.3, see also Fig. 3. From the filtered trajectories a per-pixel mean injection height (at the vent), measurement height (at each TROPOMI pixel), and flight time (i.e. the time from emission to TROPOMI pixel) were determined. Note that this approach is comparable to the approach described in Pardini et al. (2017) and Pardini et al. (2018).

2. To investigate the source of the aerosol layer that passed over RAMSES on 26-27 September 2021, backward trajectory calculations were started from the coordinates of the Lindenberg Meteorological Observatory (52.21 N, 14.12 E) in 10-minute increments between 09:00 UTC on 26 September 2021 and 09:00 UTC the next day. In contrast to the analysis described above, this approach provides time-resolved results. The HYSPLIT simulation time was set to 120 h, so that the simulations ended on 21-22 September 2021.

In addition, eight trajectories with horizontal offsets of $\pm 0.05°$ around the RAMSES site were launched to account for uncertainties in the meteorological field. The trajectories were started at heights between 1 and 8 km with a vertical increment of 50 m. Thus, 1269 back-trajectories were calculated for each time step, totaling 1,840,005 for this analysis. The trajectories were assigned to Cumbre Vieja if they passed within 100 km. From the filtered trajectories, the mean injection height (at the vent), the measurement height (at Lindenberg), and the flight time (i.e. the time from emission to 200    Lindenberg) were determined.

### 3.3  Microphysical retrieval

Measurements with RAMSES are generally not well suited for the retrieval of microphysical aerosol properties because the lidar has only one transmitter wavelength and therefore the spectral dependence of the aerosol elastic-optical properties cannot be quantified. However, as will be shown, measurement cases are an exception where the number of retrieval parameters can 205    plausibly be restricted and the particle depolarization ratio can be used as a second particle number concentration-independent measurement parameter in addition to the lidar ratio. Both conditions apply to the RAMSES measurement of the Cumbre Vieja aerosol during the night of 26-27 September, 2021.

As will be discussed in Sect. 4.4, the volcanic aerosol that reached the RAMSES site exhibited high lidar ratios ($\sim 75$ sr) and small particle depolarization ratios ($\sim 1.8$ %). Since the ambient air was dry and thus droplet formation unlikely, it can 210    be concluded that the aerosol particles were small (fine mode), solid and of non-spherical shape. Particularly, the existence of a second so-called coarse mode consisting of relatively large ash particles can be excluded. Under the conventional assumption that aerosol modes follow a lognormal distribution, this reduces the number of retrieval parameters to be determined to 3 (assuming complex refractive index and particle shape). This number corresponds to the number of independent RAMSES measurements of particle backscatter coefficient ($\beta_{\mathrm{par}}$), lidar ratio ($S_{\mathrm{par}}$) and particle depolarization ratio ($\delta_{\mathrm{par}}$), so that a re- 215    trieval becomes possible. Obviously, this requires a particle model with which the optical properties of non-spherical particles can be realistically modeled, in this case the finite-difference time-domain (FDTD) scheme is used (Yang et al., 2000). Previous analyses of polar stratospheric cloud observations demonstrated the applicability of FDTD data to aerosol measurements



with lidar (Reichardt et al., 2004, 2014). Following is a description of the retrieval method used in Sect. 4.5 to estimate the microphysical properties of the volcanic aerosol from La Palma over northern Germany.

The inversion algorithm considers a single aerosol mode with lognormal size distribution:

$$n(c) = N_{\mathrm{par}}\hat{n}(c) = \frac{N_{\mathrm{par}}}{\sqrt{2\pi}\,c\ln\sigma}\exp\left[-\frac{1}{2}\left(\frac{\ln c - \ln c_{\mathrm{m}}}{\ln\sigma}\right)^2\right],\tag{1}$$

where $N_{\mathrm{par}}$ is the total number of aerosol particles, $c$ is the half of the particle's maximum dimension, and $c_{\mathrm{m}}$ and $\sigma$ are the median half of the particle maximum dimension and the width of the distribution, respectively. Note that the independent variables do not refer to the radius of the particle, but to the semi-diameter of its longest axis. This is because the particle shape is assumed to be irregular and it is difficult to define the radius of such a particle.

The retrieval error $E$ is defined as the sum of the absolute values of the relative errors in retrieved lidar and depolarization ratios:

$$E(S_{\mathrm{par}}^{\mathrm{ret}}, \delta_{\mathrm{par}}^{\mathrm{ret}}) = \left|\frac{S_{\mathrm{par}}^{\mathrm{ret}} - S_{\mathrm{par}}^{\mathrm{mea}}}{S_{\mathrm{par}}^{\mathrm{mea}}}\right| + \left|\frac{\delta_{\mathrm{par}}^{\mathrm{ret}} - \delta_{\mathrm{par}}^{\mathrm{mea}}}{\delta_{\mathrm{par}}^{\mathrm{mea}}}\right|,\tag{2}$$

where superscripts 'mea' and 'ret' denote measured and retrieved quantities, respectively. $S_{\mathrm{par}}^{\mathrm{ret}}$ and $\delta_{\mathrm{par}}^{\mathrm{ret}}$ are calculated for a

given parameter pair $c_{\mathrm{m}}$ and $\sigma$ using the FDTD data base and following the model formulas summarized by Reichardt et al. (2002).

The objective of the inversion run is to minimize $E$ which can be regarded as a measure of the quality of a fit. First, refractive index and length-to-width (aspect) ratio of the aerosol particles are preselected. Under these assumptions the distribution parameters $c_{\mathrm{m}}$ and $\sigma$ are estimated by comparing measured and modeled $\delta_{\mathrm{par}}$ and $S_{\mathrm{par}}$ (which are independent of $N_{\mathrm{par}}$).

For this purpose, the parameter pair is varied within reasonable ranges and the inversion results are ranked according to the magnitude of the resulting retrieval error $E$. The effective half of particle maximum dimension ($c_{\mathrm{eff}}$) is computed as auxiliary output, which is defined as the ratio of distribution-averaged particle volume ($V$) and projection area ($A$):

$$c_{\mathrm{eff}} = \sum_c[\hat{n}(c)V(c)]/\sum_c[\hat{n}(c)A(c)].\tag{3}$$

To account for the variability and the statistical measurement error of measured $S_{\mathrm{par}}$ and $\delta_{\mathrm{par}}$, not the inversion run with

minimum $E$ is considered the optimal solution, but rather the average over a freely selectable number of inversion runs with the lowest $E$ values (10 in this study). The particle spectrum of the overall size distribution is therefore not necessarily lognormally distributed but a superposition of lognormal number size distributions. $N_{\mathrm{par}}$ is then calculated from the $\beta_{\mathrm{par}}$ measurement. Other bulk properties such as the aerosol surface-area concentration ($s_{\mathrm{par}}$) and the aerosol volume concentration ($v_{\mathrm{par}}$) are also obtained. By repeating the computations for different refractive indices and aspect ratios, best estimates for these quantities can

also be obtained by evaluating $E$. Finally, the plausibility and the height range of applicability of the retrieved microphysical aerosol parameters can be checked by a comparison of retrieved and measured particle extinction coefficient ($\alpha_{\mathrm{par}}$).

The FDTD data base is a compilation of theoretical optical properties. The modeled aerosol particles are assumed to be of irregular shape, i.e., they do not exhibit any symmetry axes, and to be randomly oriented in space. The calculations were performed at three wavelengths (355, 532, and 1064 nm), for five size-independent aspect ratios (0.5, 0.75, 1, 1.25, and 1.5) and for




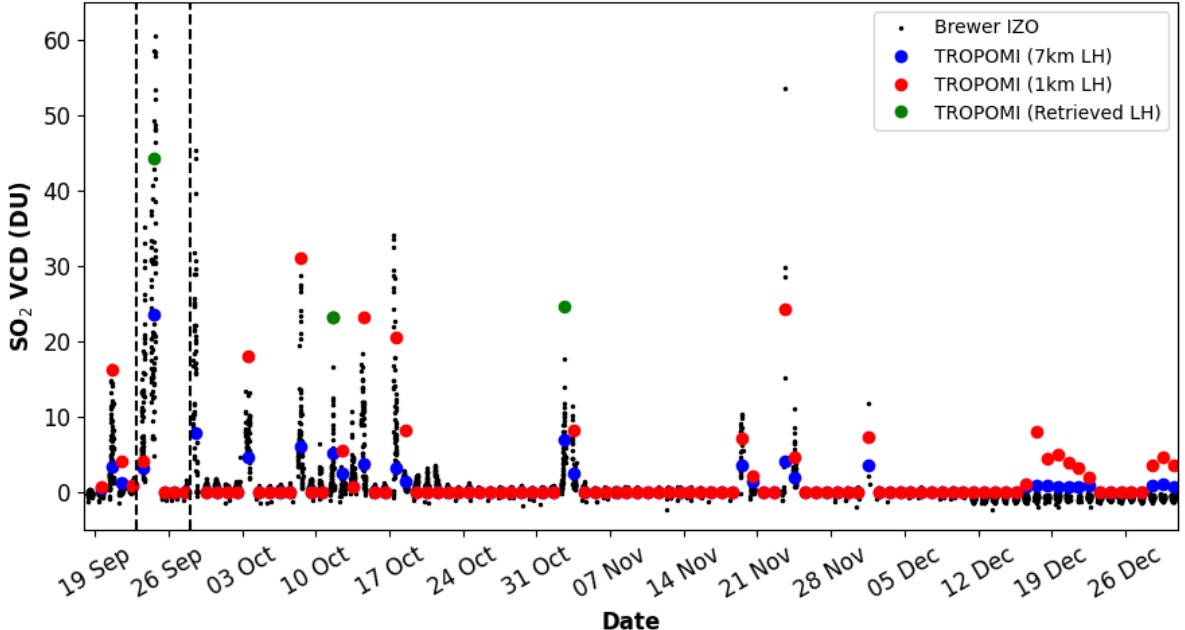

**Figure 2.** Temporal evolution of Tajogaite/Cumbre Vieja SO$_2$ VCD over the entire period of volcanic activity as measured with the ground-based IZO Brewer spectrophotometer located on Tenerife (black dots) and TROPOMI. Median TROPOMI SO$_2$ VCD are presented, calculated within a radius of 10 km around the IZO station for standardized plume heights of 1 km (red dots) and 7 km (blue dots), and for the retrieved SO$_2$ LH (green dots). Vertical dashed lines indicate the eruptive event that is analyzed in this paper in detail. Note that the Brewer instrument is located on Tenerife, hence a time offset compared to measurements on La Palma applies.

discrete particle maximum dimensions between 0.01 and 4 μm. The refractive indices cover the expected range of atmospheric particles (real part between 1.32 and 1.75, weakly absorbing), ranging from water ice over those of the Moderate-resolution Imaging Spectroradiometer (MODIS) aerosol modes (Kaufman et al., 2003) to biomass burning aerosol, soil particles, and minerals such as quartz, basalt, and volcanic ash.

## 4   Results and discussion

### 4.1   Comparison of TROPOMI and ground-based data in the vicinity of Tajogaite


Figure 2 compares SO$_2$ VCDs measured with TROPOMI and the IZO Brewer spectrophotometer on the island of Tenerife over the entire eruptive phase of the Tajogaite volcano. The agreement is satisfactory. Although TROPOMI only performs daily measurements over the Canarian Islands, it detects most of the eruptive events with high SO$_2$ amounts observed by the Brewer instrument. Around the major eruption in late September 2021, the matching can be significantly improved if an intermediate
layer height of about 4 to 5 km is assumed, which agrees well with the actual plume height evidenced by the ground-based

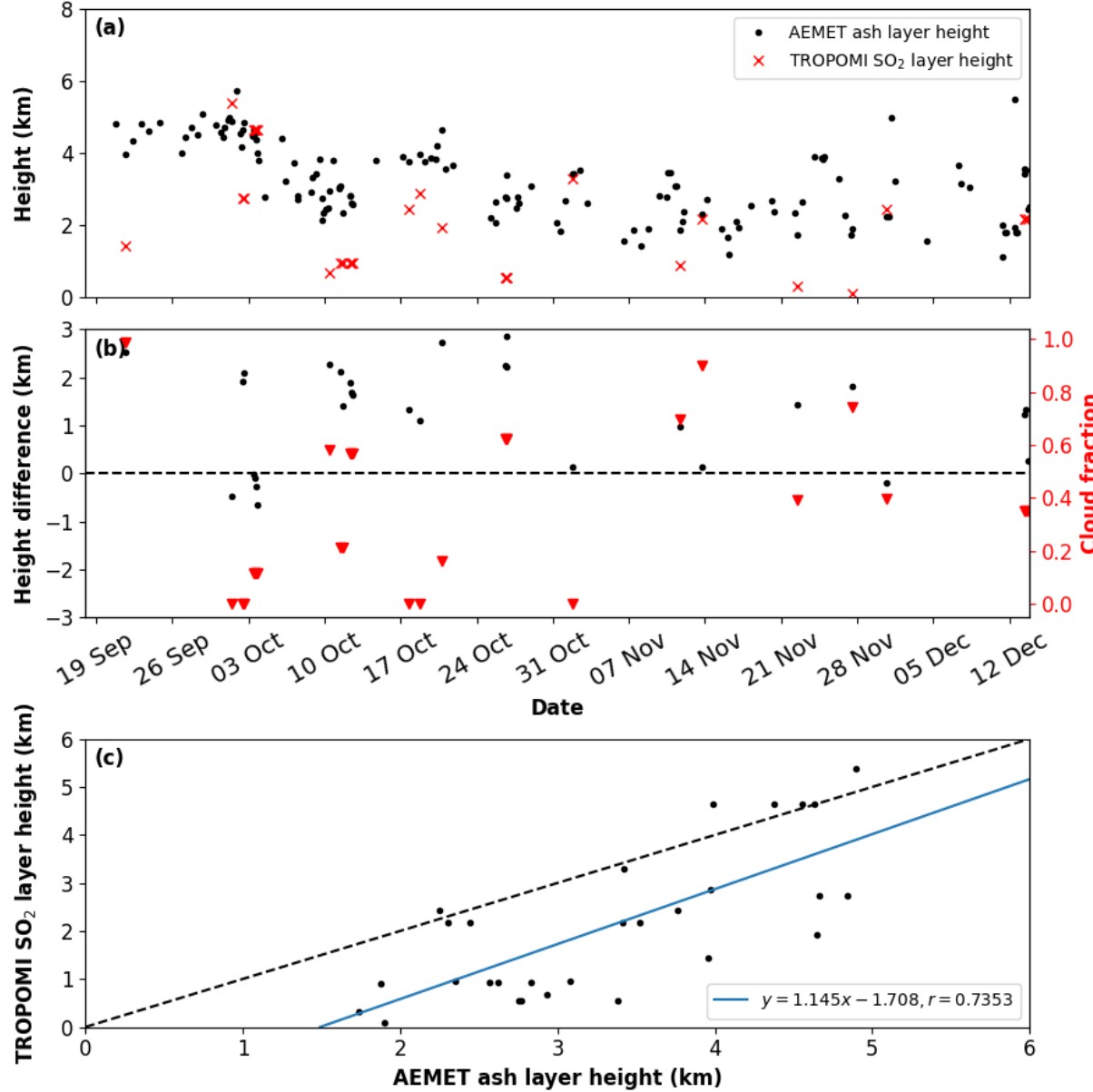

**Figure 3.** Temporal evolution of the Tajogaite plume height over La Palma between 19 September and 13 December 2021. **(a)** Plume height derived from ground-based lidar measurements at local AEMET stations (black dots) and median TROPOMI $SO_2$ layer height retrieved within a distance of 10 km (red crosses) from the corresponding station. **(b)** Difference between lidar plume height and median TROPOMI $SO_2$ layer height within 10 km radius (black dots), and TROPOMI cloud-fraction (red triangles). **(c)** Relation between lidar-derived plume height and median TROPOMI $SO_2$ layer height within 10 km radius. The result of the linear fit is shown (blue line). For reference the identity function is marked as well (dashed black line).





lidar measurements and TROPOMI $SO_2$ LH (Fig. 3a). On 12 October 2021, the TROPOMI $SO_2$ LH retrieval underestimates layer height, possibly due to the presence of volcanic ash, and hence the $SO_2$ VCD for the retrieved LH is overestimated and close to the value derived for the standard plume height of 1 km. Surprisingly, on 3 November 2021 TROPOMI $SO_2$ LH and lidar-derived plume height agree remarkably well but yet the associated TROPOMI $SO_2$ VCD is larger than the Brewer
measurement. It may be speculated that spatial inhomogeneity in the volcanic plume or some undetected clouds caused the discrepancy.

Figure 3 discusses the local long-term observations of the Tajogaite plume height. As detailed in Sect. 2.2, lidar profiles from several instruments across the island of La Palma were evaluated to determine the top height of the ash layer. The layer was highest (around 4-5 km) in the first days after the eruption, and subsided slightly after 5 October 2021 (Fig. 3a). The
final part of the eruption then again showed an explosive episode with high ash heights. Median TROPOMI $SO_2$ LH (Sect. 2.1) shows good agreement with the lidar results, even though the measurement time difference was up to 5 hours and the mean cloud-fraction over the scene was significant at times (Fig. 3b, red triangles). The differences are in the range of only 1-3 km, with a difference of lower than 1 km for cloud fractions lower than 0.5, which is quite astonishing given the dissimilar methodology. Statistically, TROPOMI $SO_2$ LH underestimates the height of the volcanic plume if compared to lidar ash height
(Fig. 3c). Although a correlation between the two data sets is evident, it is not pronounced (correlation coefficient $r = 0.74$), which is hardly surprising given the methodological difficulties discussed in detail.

### 4.2 Analysis of the northeastern Europe-bound volcanic plume

According to Milford et al. (2023), Tajogaite ejected a record daily amount of 125 kt $SO_2$ during its most intense eruption which culminated on 23 September 2021. The volcanic cloud split into a part that moved south and a part that moved northeast
towards Europe (Fig. 1), of which the latter passed over Lindenberg several days later.

Key features of the Tajogaite plume are explored for 26 September 2021, the day of the start of the RAMSES measurements. Since the retrieval of the volcanic plume height directly from the TROPOMI observations is only possible for significant levels of $SO_2$ (Sect. 2.1), an alternative method had to be devised to track the height of the increasingly diluted $SO_2$ cloud over Europe. This indirect method is based on extensive back-trajectory computations, as explained in Sect. 3.2. The criteria
described in that section were fulfilled by 6,776,014 trajectories, which corresponds to 13.84 % of the total number of started trajectories. Figures 4a-c present the mean emission height, mean measurement height, and mean plume age for the TROPOMI $SO_2$ VCD scene of 26 September 2021 (cf. Fig. 1). Every TROPOMI pixel with elevated $SO_2$ level was used as a start location for a backward trajectory. The longitudinally integrated number of trajectories reaching the volcanic vent for measurement heights between ground and 8.75 km is also depicted (Fig. 4d). Note that the number of trajectories, however, does not directly
indicate the $SO_2$ load of the trajectory. In view of the ground-based ash layer altitude measurements near the Tajogaite volcano (Fig. 3), trajectories were discarded if the height at their location closest to the vent was not within 2 to 6.5 km. This is based on the assumption that ash and $SO_2$ layers were similar.





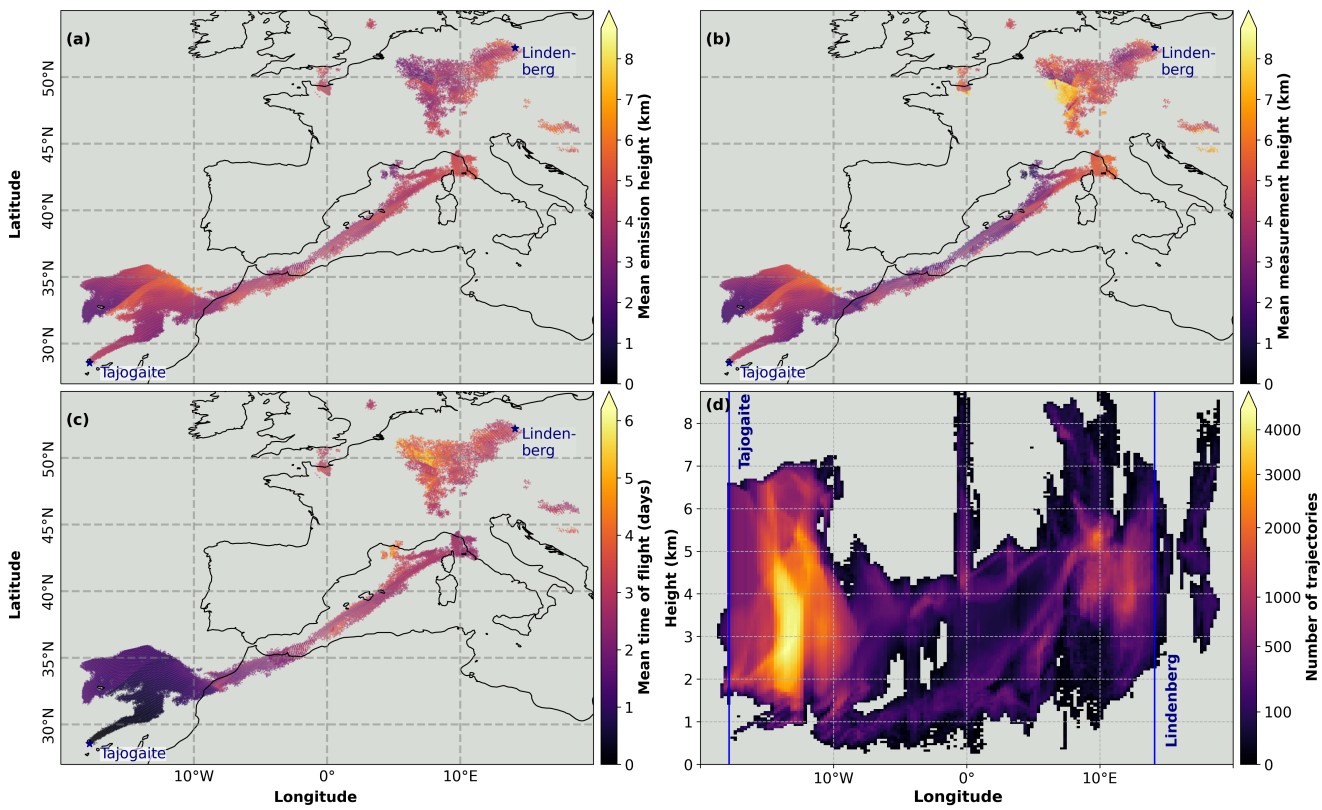

**Figure 4.** Analysis of the $SO_2$ plume over Europe on 26 September 2021 based on HYSPLIT back-trajectory calculations. For each TROPOMI pixel with enhanced $SO_2$ VCD, **(a)** mean injection height at volcanic vent, **(b)** mean layer height at measurement location, and **(c)** mean plume age are visualized; **(d)** number of trajectories reaching the vent as function of measurement height and longitude.

As expected, the volcanic $SO_2$ plume close to La Palma is relatively young, with ages of around 1 day (Fig. 4c). The age of the extended plume over the Mediterranean Sea is around 2–3 days, over Germany about 3–4 days. Thus, it can be concluded that $SO_2$ observed on 26-27 September over Germany was released from Tajogaite around 23 September 2021.

Furthermore, the plume over Germany is located around heights of 3-6 km at the time of the TROPOMI measurement with a small fraction between heights of 7 and 8 km over south-west Germany (Figs. 4b, d), and was emitted at an injection height of about 2.5-5 km (Fig. 4a). This is also visualized in Fig. 5, which shows histograms of the mean injection height at the volcanic vent (a), mean measurement height over Germany (b) and mean time of flight of the trajectories launched from each TROPOMI pixel over Germany (45-55 N, 6-15 E) that successfully arrive at the volcanic vent. Importantly, this emission height range is in excellent agreement with the direct TROPOMI $SO_2$ LH measurements on 23 September (Fig. 6b), underlining the usefulness of the backward trajectory-based analysis approach.




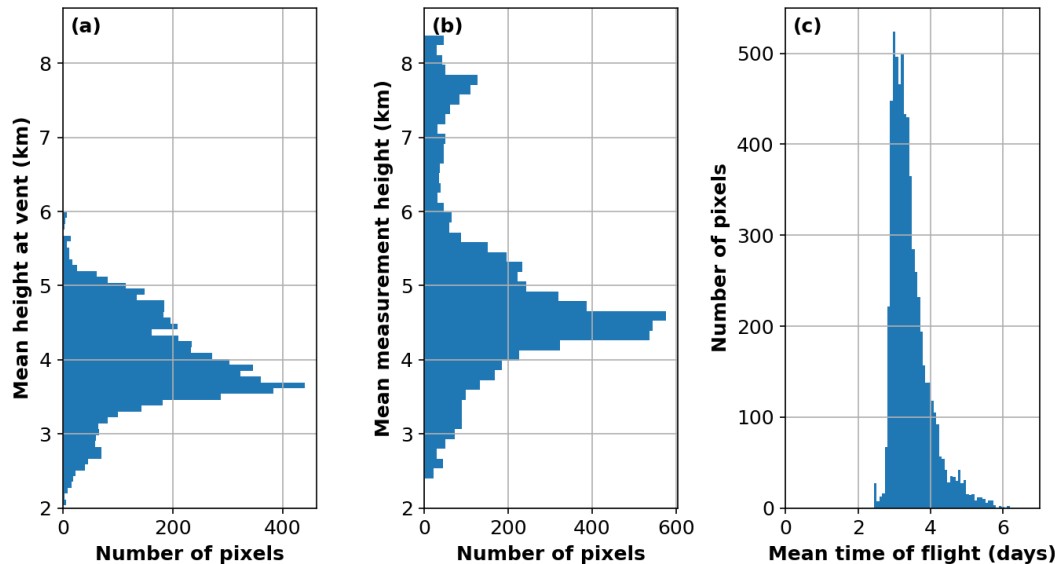

**Figure 5.** Histograms of mean trajectory properties launched from each TROPOMI pixel location over Germany (45-55 N, 6-15 E) on 26 September 2021 and arriving at the Cumbre Vieja vent based on HYSPLIT back-trajectory calculations. **(a)** Injection height at the volcanic vent, **(b)** layer height over Germany, and **(c)** Time of fligh are visualized.

## 4.3 Trajectory analysis applied to Lindenberg observations

To put the measurements of the Lindenberg Meteorological Observatory, Germany, into context with the Tajogaite eruptions, extensive trajectory calculations have been performed as described in Sect. 3.2. Figure 7 presents the results of this time-dependent trajectory analysis. Figures 7a-c show the arrival height as a function of arrival time at Lindenberg: in Fig. 7a the minimum approach distance of all trajectories to the volcano is color-coded. In Figs. 7b and 7c, however, only trajectories are considered which passed it within 100 km; in the former the emission height is color-coded, in the latter the trajectory travel time from La Palma to Lindenberg. Finally, Fig. 7d visualizes the emission heights and dates for the arrival heights of the air parcels over RAMSES.

There is a noticeable time dependence of the arrival heights in Lindenberg: until 21:00 UTC trajectories from near the volcanic vent are registered over a broad altitude range between 2 and 6 km. Then several distinct layers emerge at 2.5, around 3.5, and around 4.5 km. After 00:00 UTC only one layer persists between 4.5 and 5.5 km, slowly ascending with time.

The emission height at the volcanic vent is well correlated with the arrival height at Lindenberg (Fig. 7b). Generally, emission height increases with measurement height, albeit the temporal behavior is dynamical. For instance, between 13:00 and 21:00 UTC emission height fluctuates considerably at 3-4 km. Interestingly, plume age is connected to emission height. While between 15:00 and 16:00 UTC the travel time of the air parcels with low emission heights can reach up to 5 days, which







**Figure 6.** Measured TROPOMI $SO_2$ VCD **(a)** and layer height **(b)** of the volcanic plume at 14:57 UTC over the Canary Islands archipelago on 23 September 2021.

means that they passed by Tajogaite volcano on 21 September 2021, the earlier and later air parcels with elevated injection heights exhibit a significantly shorter transport time (Fig. 7c).



Finally, from Fig. 7d one can tell that air masses observed over Lindenberg between 3 and 4 km carried volcanic emissions
that were injected at heights between 2 and 5 km over two days starting at 09:00 UTC on 21 September 2021. For measurement altitudes above, emission height is higher and emission time confined to after 09:00 UTC on 23 September 2021. The agreement between the HYSPLIT simulations and the RAMSES lidar measurements (Fig. 8) is quite satisfying, especially the fluorescent layers at 3.2 and 4.5 km are well reproduced by the HYSPLIT simulations between 22:00 and 00:00 UTC.

## 325   4.4   Lindenberg Meteorological Observatory: volcanic aerosol optical properties

In September 2021, RAMSES measured continuously first, but starting on September 24 operation was restricted to night mode due to generally unfavorable weather conditions. Therefore, the measurement of the Tajogaite aerosol on September 26-27 only started at 17:30 UTC. Incidentally, measurements during cloud gaps in the nights before and after provide no indication of the presence of volcanic aerosol, which is in agreement with the TROPOMI $SO_2$ measurements (Fig. 1). The RAMSES dataset

extended by ceilometer measurements on site in the early afternoon of 26 September and morning of 27 September provides the end times of the extensive back-trajectory analyses in Sect. 4.3.

    Figure 8 depicts the temporal evolution of the aerosol event. In addition to the more common measurement quantities such as particle backscatter coefficient, lidar and particle depolarization ratio and relative humidity, those that characterize the fluorescence of the Tajogaite aerosol are also shown. These are the fluorescence backscatter coefficient as the sum of the

spectral backscatter coefficients ($\beta^{FL}$) from 455 to 535 nm, which is therefore assigned the false color cyan (Fig. 8c), and the spectral fluorescence capacity in the same wavelength range ($C_{cyan}^{FL}$, Fig. 8e), which is defined as the quotient of mean $\beta^{FL}$ and the (elastic) particle backscatter coefficient (Reichardt, 2014; Reichardt et al., 2023). The latter is a measure of how strongly the observed aerosol fluoresces and, together with the knowledge of the shape of the fluorescence spectrum, allows one to deduce the aerosol type. In this night, clouds only appear towards the end of the observation period (discernible by the red and

deep blue patches in the $\beta_{par}$ and $S_{par}$ displays, respectively), so that the data quality is excellent.

    The aerosol field showed relatively little dynamics during the entire measurement night. Three layers can be identified, which differ significantly in their aerosol properties. The layer below approx. 2.4 km exhibits relatively low $\beta_{par}$ and $S_{par}$, at times significant $\delta_{par}$, and comparatively high fluorescence backscatter coefficients and spectral fluorescence capacity.

    Clearly delineated by its properties, the layer that will be the main focus of this study is located above in the height range

of approximately 2.5 to 4 km. This filament can be traced back with a high degree of certainty to the eruptive phase of the Tajogaite volcano on 23 September 2021 (see Sect. 4.3) and at the same time exhibits homogeneous elastic and inelastic optical properties that can be measured with low statistical errors. In contrast to the bottom layer, it is characterized by extremely low $\delta_{par}$ and $C_{cyan}^{FL}$, low fluorescence backscatter coefficient, elevated $\beta_{par}$ and high $S_{par}$. The dynamic evolution of $\beta_{par}$ is not reflected in $S_{par}$ and $\delta_{par}$ and is therefore caused by variations in particle concentration and not in particle type. For the

discussion that follows it is also important to note that the relative humidity is generally low.

    Above an altitude of 4 km, a third aerosol layer appears around 21 UTC (Fig. 8, panels b, c, and e), which probably also consists of volcanic aerosol but is so optically thin that it is hardly visible in $\beta_{par}$ and $S_{par}$ cannot be determined. A prominent feature is its elevated depolarization ratio. Relative humidity is so high that clouds can form sporadically.





**Figure 7.** Analysis of air parcel history over Lindenberg Meteorological Observatory for the time period between 09:00 UTC on 26 September 2021 and 09:00 UTC on 27 September 2021 based on HYSPLIT back-trajectory calculations. Trajectory start heights ranged from 1 to 8 km. **(a)** Minimum approach distance to Tajogaite , **(b)** height at minimum distance to Tajogaite for approach distances < 100 km, and **(c)** air-parcel travel time between Tajogaite and Lindenberg. Vertical red lines indicate the RAMSES lidar measurement times, which are analyzed in Sect. 4.5. From the latter two datasets, **(d)** the emission heights and dates for the arrival heights of the air parcels over Lindenberg can be derived.





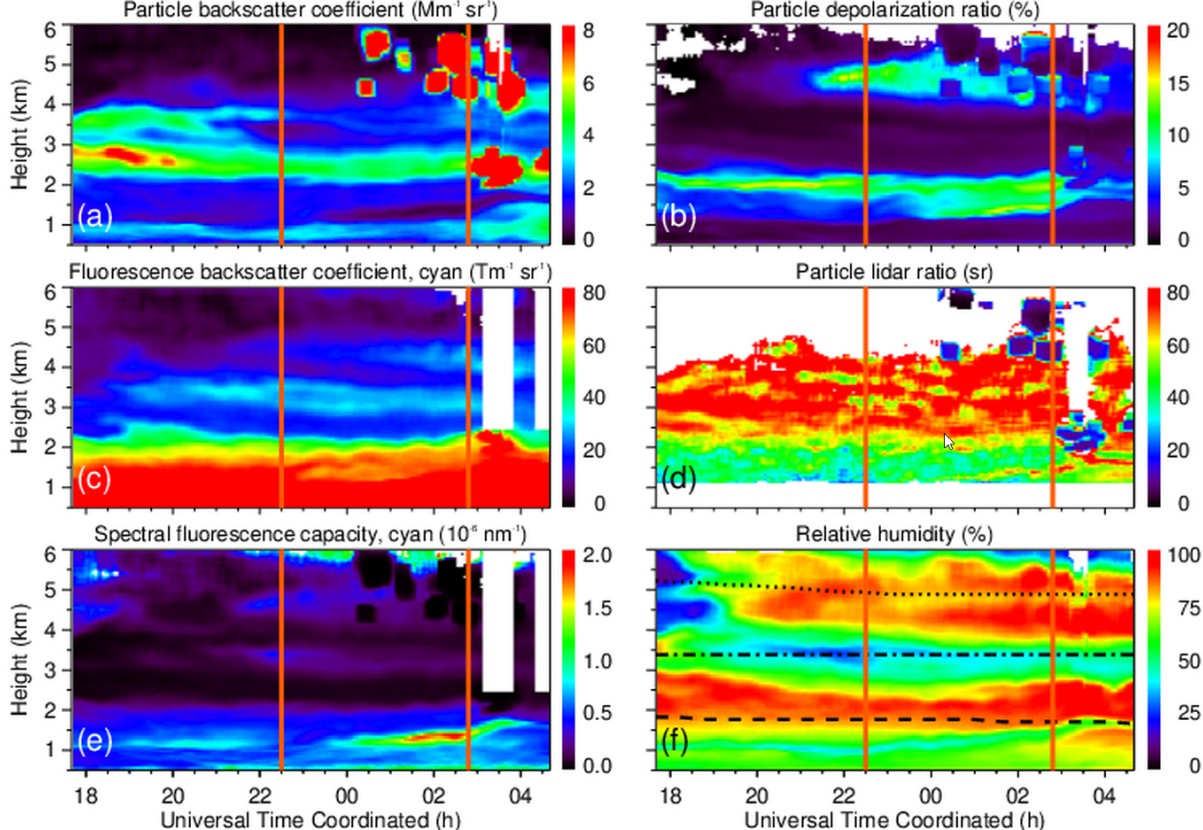

**Figure 8.** Temporal evolution of **(a)** particle backscatter coefficient, **(b)** particle depolarization ratio, **(c)** fluorescence backscatter coefficient (cyan false color: spectrum integrated from 455 to 535 nm), **(d)** particle lidar ratio (not corrected for multiple scattering), **(e)** spectral fluorescence capacity (mean value, 455-535 nm), and **(f)** relative humidity (with respect to water; 10 °C, 0 °C and -10 °C isotherms indicated by black lines) as measured with RAMSES in the night of 26-27 September 2021 between 17:40 and 04:40 UTC. Measurements at 22:30 and 02:48 UTC are analyzed in detail in Fig. 9 (times marked by vertical orange lines). For each profile, 1200 s of lidar data are integrated, the calculation step width is 120 s. The resolution of the raw data is 60 m, signal profiles are smoothed with a sliding-average length increasing with height. White areas indicate where data were rejected by the automated quality control process.

For a quantitative discussion, the profile measurements at 22:30 and 02:48 UTC are depicted in Fig. 9. It can be clearly seen
that the aerosol optical properties, both elastic and inelastic, remained essentially unchanged over the measurement period, and the vertical stratification was also maintained. Only an increase in relative humidity can be observed, which ultimately led to a swelling of the aerosols and to cloud formation, particularly at heights above 4 km, as manifested by the $\beta_{\mathrm{par}}$ profile (Fig. 9a).

The aerosol layer between 2.5 and 4 km remained dry at its center but got moister towards its bounds (up to 90 % relative humidity). An important observation is that $S_{\mathrm{par}}$ and $\delta_{\mathrm{par}}$ did not respond to these changes in relative humidity which indicates
that the particles were either insoluble in water or had a high deliquescence relative humidity (e.g., ammonium sulfate exhibits a

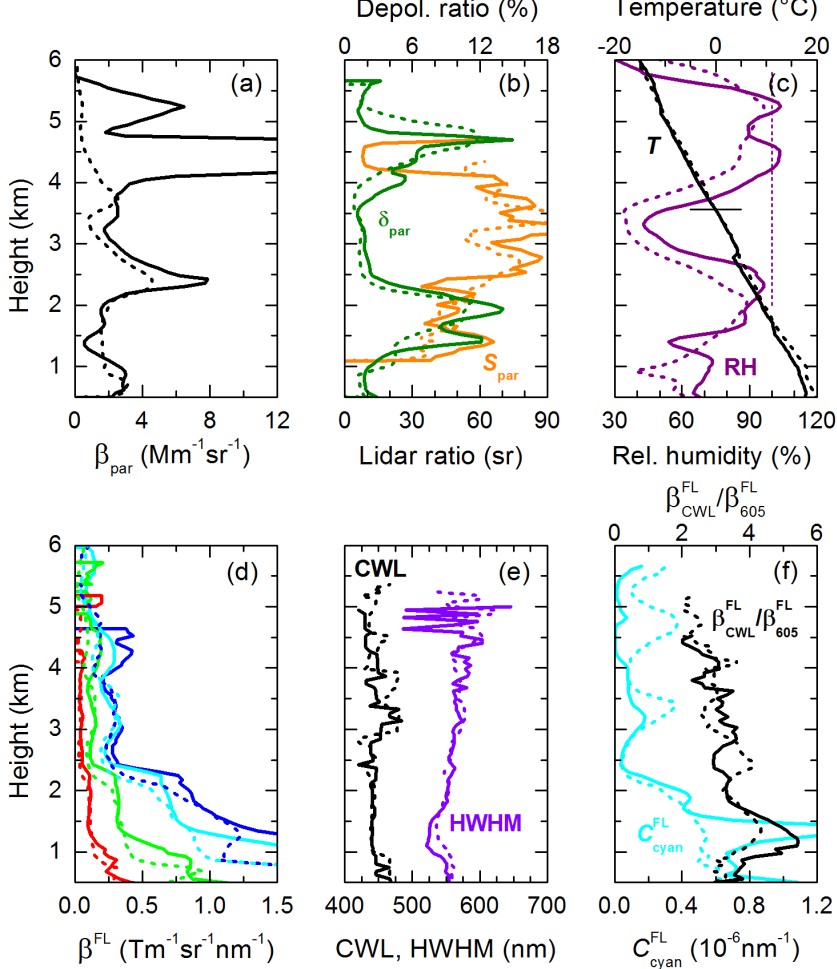

**Figure 9.** RAMSES measurements in the night of 26-27 September 2021 at 22:30 UTC (dashed curves) and 02:48 UTC (solid curves). Profiles of **(a)** particle backscatter coefficient ($\beta_{par}$), **(b)** particle lidar ratio ($S_{par}$) and particle depolarization ratio ($\delta_{par}$), **(c)** relative humidity (RH, with respect to water and ice above and below 0 °C, respectively) and temperature ($T$), **(d)** mean spectral fluorescence backscatter coefficient ($\beta^{FL}$) at red, green, cyan and blue wavelengths, **(e)** wavelength of the spectral maximum (CWL) and of the half-maximum value on the long-wavelength shoulder of the fluorescence spectrum (HWHM), and **(f)** spectral fluorescence capacity at cyan wavelengths ($C^{FL}_{cyan}$) and ratio of spectral fluorescence backscatter coefficients at the maximum of the fluorescence spectrum and at 605 nm ($\beta^{FL}_{CWL}/\beta^{FL}_{605}$). Error bars have been omitted for conciseness.

deliquescence relative humidity of 81.9 % at 4 °C, Brooks et al., 2002). Furthermore, the minimum relative humidity of around 34 % observed at 22:30 UTC is below the efflorescence relative humidity of many atmospheric compounds such as sulfates and nitrates (Peng et al., 2022), and so if these aerosol particles had been droplets previously, they would have solidified due to water loss by the time of measurement. It can therefore be assumed that the volcanic aerosol particles were not of spherical shape. Instead, the low $\delta_{par}$ values near 2 % and the high $S_{par}$ values around 75 sr (Fig. 9b) must be attributed to scattering by




solid particles. These particles are definitely not volcanic ash, because lidar measurements during several volcanic eruptions provided ash $\delta_{par}$ and $S_{par}$ values of $> 30$ % and $< 65$ sr, respectively, in the troposphere (Ansmann et al., 2010; Groß et al., 2012; Pisani et al., 2012). Saharan dust can also be excluded for the same reason (Groß et al., 2012).

A further indication of the aerosol type of the central layer is provided by the fluorescence properties, which are illustrated in Fig. 9, panels d-f. It should be noted at this point that the fluorescence spectrum cannot originate from $SO_2$ itself but must come from the aerosol particles. An estimation shows that even under favorable experimental and atmospheric assumptions, the fluorescence backscatter signal of $SO_2$ could explain at most 0.1 % of the measured count rates in the cyan spectral range, because the absorption cross section of $SO_2$ at 355 nm ($\sim 10^{-22}$ cm$^2$ molecule$^{-1}$, Manatt and Lane, 1993), the $SO_2$ column value measured above Lindenberg (4 DU) and the fluorescence backscatter coefficients between 455 and 535 nm ($< 35$ Tm$^{-1}$sr$^{-1}$, Fig. 8c) are simply too small. The wavelengths of the maximum of the fluorescence spectrum and of the half-maximum value on its long-wavelength shoulder (Fig. 9e) as well as the ratio of the spectral backscatter coefficients at the maximum of the fluorescence spectrum and at 605 nm (Fig. 9f) characterize the shape of the fluorescence spectrum. Values of 440 nm, 560 nm, and around 3, respectively, and $C_{cyan}^{FL}$ values $< 4 \times 10^{-7}$ nm$^{-1}$ are similar to those measured in the boundary layer under undisturbed conditions (no domestic fires, no forest fires) and differ significantly from the spectra of biomass burning aerosol (BBA) frequently measured with RAMSES in the troposphere and lower stratosphere (Reichardt et al., 2023). For example, the maximum of the spectrum for BBA is well above 500 nm and $C_{cyan}^{FL}$ is often more than ten times higher.

In summary, spectral fluorescence properties and capacity indicate the presence of inorganic aerosols, and particle elastic optical properties suggest that the particles were small and of irregular shape. If any volcanic ash had been injected into the air masses that were monitored in the central layer, by the time the plume reached Lindenberg the particles would have disappeared. Under these assumptions a microphysical interpretation of the RAMSES measurements is presented in Sect. 4.5.

Interestingly, the adjacent aerosol layers show significantly higher $\delta_{par}$ and lower $S_{par}$. Between 1.4 and 2.4 km, maximum $\delta_{par}$ values at both measurement times are $> 10$ % and $S_{par}$ fluctuates around 50 sr. The spectral fluorescence properties are little changed, only $C_{cyan}^{FL}$ increases slightly. Above 4 km some of the measurement quantities are not available because either the aerosol layer is optically too thin or masked by the cloud, but at 22:30 UTC comparable $\delta_{par}$ values can be measured. These observations would be consistent with the assumption that there could have been admixtures of mineral aerosol such as volcanic ash or Saharan dust. According to the back-trajectory analyses in Sect. 4.3, this is entirely plausible.

## 4.5 Lindenberg Meteorological Observatory: volcanic aerosol microphysical properties

The analysis of the particle optical properties in Sect. 4.4 led to the conclusion that the volcanic aerosol layer between 2.5 and 4 km consisted of inorganic, small, solid and irregularly shaped particles. In the following, an attempt is made to determine some of the microphysical properties of these — in the terminology of Córdoba-Jabonero et al. (2023) — non-ash particles.

The retrieval scheme described in Sect. 3.3 was employed, with the retrieval runs configured as follows. The particle spectrum is considered monomodal. To determine the median half of the particle maximum dimension and the width of the normalized size distribution ($c_m$ and $\sigma$), $S_{par}$ and $\delta_{par}$ layer mean values of 75 sr and 1.8 %, respectively, are assumed as $S_{par}^{mea}$





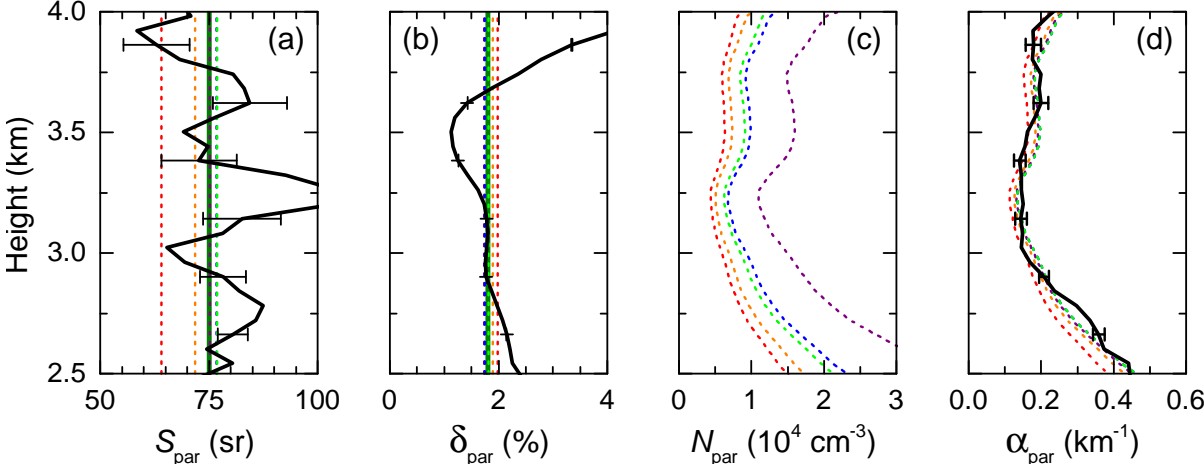

**Figure 10.** Retrieval of the microphysical properties of the volcanic aerosol layer measured with RAMSES between 2.5 and 4 km on 27 September 2021 at 02:48 UTC. Profiles of **(a)** measured (black curve, with statistical-error bars), measured mean (dark green vertical line) and retrieved (dashed colored vertical lines) particle lidar ratios ($S_{\mathrm{par}}$), **(b)** measured, measured mean and retrieved particle depolarization ratios ($\delta_{\mathrm{par}}$), **(c)** retrieved particle number concentrations ($N_{\mathrm{par}}$), and **(d)** measured and retrieved particle extinction coefficients ($\alpha_{\mathrm{par}}$). Measured mean $S_{\mathrm{par}}$ and $\delta_{\mathrm{par}}$ were used as input to the inversion algorithm to estimate the defining parameters of the normalized size distribution assumed to be of monomodal lognormal shape. Retrieval results presented are the mean values over the 10 inversion runs with lowest retrieval error obtained for particle complex refractive indexes of $1.40 - i0.002$ (violet), $1.45 - i0.0035$ (blue), $1.51 - i0.000$ (green), $1.55 - i0.005$ (orange), and $1.65 - i0.005$ (red), respectively, and an aspect ratio of 1. Measured particle backscatter coefficients were then used to retrieve the associated $N_{\mathrm{par}}$ profiles. From the retrieval results $\alpha_{\mathrm{par}}$ can be calculated, estimated and measured $\alpha_{\mathrm{par}}$ profiles are compared in panel **(d)**. Retrieved size distributions are shown in Fig. 12.

and $\delta_{\mathrm{par}}^{\mathrm{mea}}$ (see Figs. 10a, b). These values are taken from the RAMSES measurement at 02:48 UTC, but can be considered representative for the earlier measurement as well (see Fig. 9). Aspect ratio (shape) and refractive index are prescribed for each run, where $c_{\mathrm{m}}$ is varied between 0.05 and 0.20 μm (step size of 0.01 μm) and $\ln \sigma$ is varied between 0.10 and 0.45 (step size of 0.01). The individual results are ordered according to $E$. However, in order to account better for measurement errors and natural variability, not the pair of values $c_{\mathrm{m}}$ and $\sigma$ with minimum retrieval error is considered the optimal solution, but rather

the mean value from the 10 best retrieval runs. The resulting normalized particle size distribution is therefore a superposition of lognormal distributions and thus no longer a normal distribution itself. The aspect ratio is varied over 5 discrete values between 0.50 (plate-like particles) and 1.5 (elongated particles).

The refractive index is also varied over 5 values from 1.4 to 1.65 (real part). This is intended to obtain an indication of the aerosol type. For example, it is conceivable, albeit unlikely, that the aerosol mode consisted of ultra-fine ash particles. Both real

and imaginary parts of the refractive index of volcanic ash increase when the silicon dioxide ($SiO_2$) mass fraction decreases. Basalt (49 % $SiO_2$), for instance, exhibits a refractive index of about $1.625 - i0.0017$ at 355 nm (Vogel et al., 2017). Because Cumbre Vieja lava has a low $SiO_2$ content, it is therefore worthwhile to carry out retrieval runs with a refractive index in the





**Table 2.** Summary of retrieval results obtained for the volcanic aerosol layer between 2.5 and 4 km measured on 27 September 2021 at 02:48 UTC.*

| Refractive index | $c_m$ (µm) | $\ln\sigma$ | $S_{par}^{ret}$ (sr) | $\delta_{par}^{ret}$ (%) | $E$ (%) | $\overline{E}$ (%) | $\hat{\beta}_{par}$ (µm² sr⁻¹) | $\hat{\alpha}_{par}$ (µm²) | $c_{eff}$ (µm) |
|---|---|---|---|---|---|---|---|---|---|
| $1.40 - i0.002$ | 0.10 | 0.38 | 76.0 | 1.76 | 3.3 | 11.8 | 0.000169 | 0.01285 | 0.100 |
| | 0.11 | 0.35 | 81.4 | 1.80 | 8.7 | | 0.000197 | 0.01605 | 0.104 |
| | 0.09 | 0.42 | 73.8 | 1.93 | 9.0 | | 0.000145 | 0.01071 | 0.097 |
| $1.45 - i0.0035$ | 0.12 | 0.28 | 76.6 | 1.74 | 5.3 | 10.2 | 0.000266 | 0.02039 | 0.102 |
| | 0.11 | 0.32 | 71.8 | 1.76 | 6.2 | | 0.000239 | 0.01718 | 0.099 |
| | 0.12 | 0.29 | 78.3 | 1.86 | 7.5 | | 0.000273 | 0.02140 | 0.103 |
| $1.51 - i0.000$ | 0.13 | 0.11 | 75.2 | 1.82 | 1.1 | 3.6 | 0.000261 | 0.01960 | 0.093 |
| | 0.13 | 0.10 | 74.1 | 1.81 | 1.6 | | 0.000258 | 0.01912 | 0.093 |
| | 0.12 | 0.26 | 73.8 | 1.81 | 2.3 | | 0.000316 | 0.02333 | 0.099 |
| $1.55 - i0.005$ | 0.12 | 0.23 | 69.4 | 1.83 | 9.2 | 15.8 | 0.000353 | 0.02454 | 0.095 |
| | 0.12 | 0.22 | 68.2 | 1.77 | 10.8 | | 0.000346 | 0.02358 | 0.094 |
| | 0.12 | 0.24 | 70.6 | 1.91 | 12.0 | | 0.000362 | 0.02556 | 0.097 |
| $1.65 - i0.005$ | 0.12 | 0.17 | 67.3 | 2.03 | 23.0 | 24.4 | 0.000413 | 0.02780 | 0.090 |
| | 0.12 | 0.16 | 66.3 | 2.01 | 23.2 | | 0.000406 | 0.02693 | 0.089 |
| | 0.12 | 0.18 | 68.1 | 2.06 | 23.4 | | 0.000422 | 0.02873 | 0.091 |

*The measured lidar and depolarization ratios representative of the layer to be emulated are 75 sr and 1.8 %, respectively. The particles are assumed to have an irregular shape with an aspect ratio of 1 and a lognormal size distribution. For each refractive index, the results of the three best retrieval runs are presented; see text for details.

$c_m, \sigma$ – retrieved size distribution parameters: median half of particle maximum dimension and standard deviation; $S_{par}^{ret}, \delta_{par}^{ret}$ – retrieved lidar and depolarization ratio; $E$ – individual retrieval error (sum of the absolute values of the relative errors in retrieved lidar and depolarization ratios); $\overline{E}$ – retrieval error averaged over the 10 best retrieval runs; $\hat{\beta}_{par}, \hat{\alpha}_{par}$ – retrieved backscatter and extinction coefficient of a single particle, distribution-averaged; $c_{eff}$ – effective half of particle maximum dimension (ratio of distribution-averaged particle volume and projection area).

range of 1.55 to 1.65 (real part). Moreover, the real refractive index of soil particles shows values of 1.42 to 1.73 in the visible and near-infrared light spectrum, with most data between 1.5 and 1.7 (Ishida et al., 1991). Finally, Ebert et al. (2002) published

secondary electron images and summarized refractive indexes of airborne particles. Note that in the following this paper only refers to the real part of the refractive index. According to these data, one would expect a rather isometric shape (aspect ratio around 1) and a refractive index between 1.5 and 1.53.

Table 2 summarizes some retrieval results obtained for isometric aerosol particles. As examples the output of the three individual retrieval runs with minimum $E$ are listed for each refractive index, the average error $\overline{E}$ of the 10 best runs is also

given. Optimum results are obtained for particles with an refractive index of 1.51. Retrieval errors generally increase at both



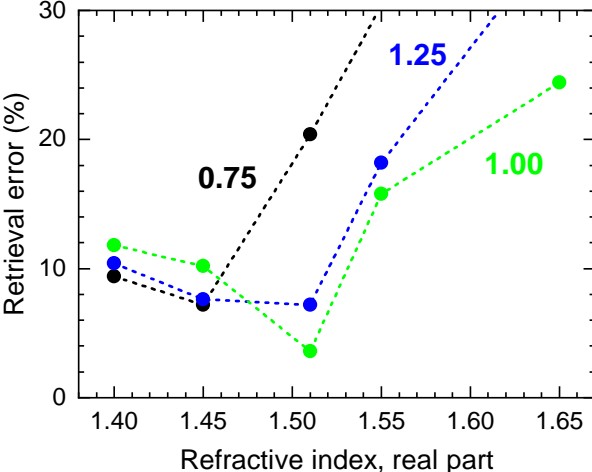

**Figure 11.** Dependence of the retrieval error of the optimum solution ($\overline{E}$) on refractive index and aspect ratios of 0.75, 1.00, and 1.25.

smaller and larger refractive indexes, and so do the differences between measured and retrieved aerosol optical properties. Distribution-averaged particle backscatter and extinction coefficients ($\hat{\beta}_{\mathrm{par}}$, $\hat{\alpha}_{\mathrm{par}}$) exhibit a significant trend with refractive index, while the effective half of particle maximum dimension ($c_{\mathrm{eff}}$) is relatively invariant ($< 20\,\%$ variation).

Figure 11 visualizes the dependence of the retrieval error of the optimum solution ($\overline{E}$) on refractive index and aspect ratio.
With increasing asphericity, the minimum of $\overline{E}$ shifts to smaller refractive indexes. For all aspect ratios considered, $\overline{E}$ is much larger for refractive indexes $> 1.51$, so it can be ruled out with some certainty that the aerosol layer over Lindenberg consisted of fine volcanic ash or optically similar material.

As detailed in Sect. 3.3, the retrieval results can be used to calculate the particle number concentration ($N_{\mathrm{par}}$) from the particle backscatter coefficient ($\beta_{\mathrm{par}}$) and then retrieve the modeled particle extinction coefficient ($\alpha_{\mathrm{par}}$). Figure 10c depicts the
$N_{\mathrm{par}}$ results obtained for the optimum solutions (averages over the 10 best retrieval runs) for all refractive indexes considered (aspect ratio of 1), and Fig. 10d compares retrieved and measured $\alpha_{\mathrm{par}}$ profiles. The underlying particle size distributions are shown in Fig. 12. $N_{\mathrm{par}}$ profiles are relatively close to one another, with the exception of the profile with the lowest refractive index, where $N_{\mathrm{par}}$ is significantly larger, which can be explained by the increased contributions of small particles to the size distribution (Fig. 12). This discrepancy disappears when the retrieved extinction coefficient is considered, because
a higher number is compensated with smaller projection areas of the particles ($\hat{\alpha}_{\mathrm{par}}$ decreases with refractive index, Tab. 2). Accordingly, all modeled extinction profiles are within or close to the statistical error limits of the RAMSES measurement. The two $\alpha_{\mathrm{par}}$ profiles obtained with the largest refractive indexes show the worst agreement, which again suggests that the particles actually had a refractive index $< 1.55$. The comparison between measured and modeled extinction coefficients thus provides an opportunity to estimate the height range of the measurement in which the retrieval can be considered representative. It is
therefore not surprising that the $\alpha_{\mathrm{par}}$ profiles show large discrepancies above $4\,\mathrm{km}$ in the layer with cloud formation and below $2.5\,\mathrm{km}$ in the aerosol layer with decreased $S_{\mathrm{par}}$ and increased $\delta_{\mathrm{par}}$ (not shown).

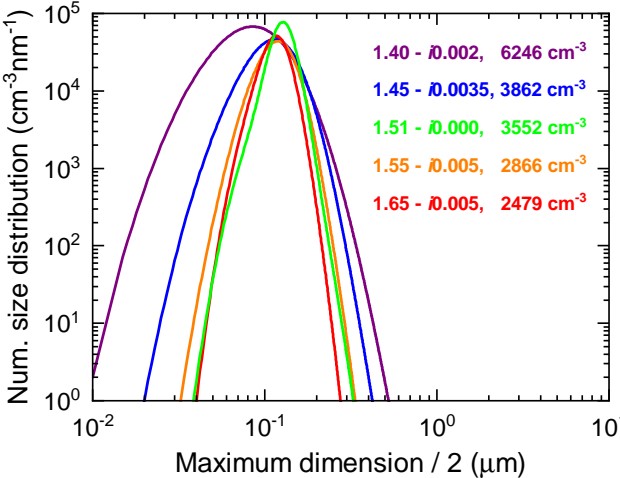

**Figure 12.** Size distributions retrieved for the volcanic aerosol layer between 2.5 and 4 km measured on 27 September 2021 at 02:48 UTC. Curves display averages over the 10 best retrieval runs for each refractive index. Because the particle shape is irregular, it is difficult to define the radius of such particle. Therefore, the results are presented as a function of the half of the particle maximum dimension. Assumed refractive index and retrieved particle number concentration ($\beta_\mathrm{par} = 1\ \mathrm{Mm}^{-1}\,\mathrm{sr}^{-1}$) of the aerosol modes are indicated. The aspect ratio is 1.

It should be noted that even a modeling attempt specifically tailored to the lower layer did not lead to satisfactory results. The solution space was very limited and the retrieval errors were high. The characterizing optical properties ($S_\mathrm{par} = 50$ sr, $\delta_\mathrm{par} = 11$ %, Fig. 9) cannot be reproduced under the assumption of a monomodal particle size distribution. This finding

supports the assumption made in Sect. 4.4 that larger particles could also have been present in this layer, forming a coarse mode. A bimodal approach would therefore be required to retrieve its microphysical properties.

For the non-ash particles of the volcanic aerosol filament between 2.5 and 4 km a volume concentration ($v_\mathrm{par}$) of ($10.6\pm0.5$) $\mathrm{\mu m}^3\,\mathrm{cm}^{-3}$ per $\beta_\mathrm{par} = 1\ \mathrm{Mm}^{-1}\mathrm{sr}^{-1}$ can be found for the refractive index of 1.51. With a refractive index of 1.45, $v_\mathrm{par}$ is 9 % larger. To calculate the aerosol mass concentration, the mass density of the particles must be assumed. Córdoba-Jabonero

et al. (2023) use a mass density of 1.5 $\mathrm{g\,cm}^{-3}$ for non-ash particles that constitute the fine aerosol mode, Ebert et al. (2002) summarize values between 1.6 and 2.3 $\mathrm{g\,cm}^{-3}$ for different types of sulfates. Applying this range of densities and considering the $\beta_\mathrm{par}$ profile between 2.5 and 4 km, mass column values $M$ between 70 and 110 $\mathrm{mg\,m}^{-2}$ can be obtained. Mean mass concentration amounts to 45-70 $\mathrm{\mu g\,m}^{-3}$. With an optical depth (OD) of 0.33, the mean mass conversion factor $f = M/\mathrm{OD}$ ranges between 0.21 and 0.33 $\mathrm{g\,m}^{-2}$ at 355 nm. Córdoba-Jabonero et al. (2023) retrieved a similar $f$ value for non-ash particles

($\sim 0.31\ \mathrm{g\,m}^{-2}$), which is quite remarkable because completely different retrieval schemes were used in the two studies, and the lidar wavelengths were different (355 vs. 532 nm).

Finally, the question arises as to whether the type of volcanic aerosol observed over Lindenberg can be inferred based on the measurements and the retrieval results. As explained in Sect. 4.4, organic aerosols and aerosols with large mineral particles can be excluded based on the optical properties. Furthermore, despite all its limitations, the results of the retrieval suggest that



the non-ash particles had a solid, isometric to slightly plate-like irregular shape with an effective half of particle maximum dimension around 0.1 μm and a refractive index of about 1.51 (1.45 with lower probability). Moreover, the lack of sensitivity of the particle properties to wide variations in relative humidity implies that they were poorly soluble in water.

Emissions from volcanoes generally encompass gases like $SO_2$ and particles. Volcanic ash, if ejected at all, was not observed by RAMSES during its 26-27 September 2021 operation. Other particles include primary sulfates, a term which is used to
differentiate between sulfates directly emitted and those produced in the atmosphere (secondary sulfates, Martin et al., 2014). According to Allen et al. (2002), primary sulfates are present mainly in the form of sulfuric acid droplets and are subject to evaporation in dry air. $SO_2$ can undergo different reaction paths in the atmosphere that may lead to the formation of particulate (secondary) sulfate (Mather et al., 2003). In gas-phase homogeneous reactions the concentration of the hydroxyl radical is paramount. The $SO_2$ conversion rates are generally slow and depend on daylight, temperature and humidity (around 5–10
$\% \, h^{-1}$ of $SO_2$ reacted in summer and 0.3-1 $\% \, h^{-1}$ of $SO_2$ reacted in winter), Pattantyus et al. (2018) report even smaller conversion rates (daytime: 0.8-5 $\% \, h^{-1}$; nighttime: 0.01-0.07 $\% \, h^{-1}$). Aqueous-phase reactions are much faster (20-100 $\% \, h^{-1}$ of $SO_2$ reacted), but require high humidity and water droplets (Mather et al., 2003).

In the case of the RAMSES observations, aqueous-phase reactions could not have played a dominant role. If cloud processing had occurred, all $SO_2$ would have been consumed by the time the northeastern plume reached Lindenberg. Especially the
altitude range between 2.5 and 4 km was too dry. So possibly RAMSES observed in this layer secondary sulfate aerosol which was produced by gas-phase homogeneous reactions. Associated $SO_2$ conversion rates may have been so slow that even after several days of atmospheric transport, a significant amount of $SO_2$ could still be observed. The performed sensitivity test is supportive of this hypothesis, because the absolute minimum in $\overline{E}$ is found for isometric particles with a refractive index of 1.51, which is in accordance with the study of Ebert et al. (2002) on atmospheric sulfate particles, and the retrieved mass
conversion factor is comparable to the one published by Córdoba-Jabonero et al. (2023) for non-ash particles. But still, it remains unknown how much of the aerosol measured with RAMSES was actually volcanic secondary sulfate since other solid particles with a similar refractive index may have contributed as well.

Incidentally, Gebauer et al. (2023) arrive at a similar conclusion regarding their measurements of planetary boundary layer aerosols when the southbound plume of Cumbre Vieja crossed over Mindelo, Cape Verde, in the morning of 24 September
2021. According to Gebauer et al. (2023), sulfate aerosol was observed. A distinctive difference to the RAMSES measurements is, however, that aerosol $\delta_{par}$ was much lower (layer mean value of $0.4 \pm 0.3$ % at 355 nm), and so the phase was likely different. The boundary-layer particles were probably droplets rather than dry solid sulfates, which is reasonable because they were exposed to high humidity over the Atlantic ocean and thus aqueous-phase reactions dominated. The significant aerosol extinction coefficient of 0.8 $km^{-1}$ (four times higher than measured in the sulfate layer over Lindenberg) may be indicative for
water uptake by the particles.



## 5    Summary and conclusions

The Tajogaite volcano on the western flank of the Cumbre Vieja ridge on La Palma, Canary Islands, was volcanological active between 19 September and 25 December 2021. The focus of this publication has been its first and strongest eruptive event, which occurred on 22-23 September 2021 and resulted in $SO_2$ emissions of more than 10 DU. The northeastern leg of the asso-

ciated plume headed towards central Europe where it was observed with the high-performance spectrometric fluorescence and Raman lidar RAMSES when it passed over the Lindenberg Meteorological Observatory of the German Meteorological Service near Berlin, Germany, in the night of 26-27 September 2021. For the first time, satellite and ground-based measurements have been combined to characterize this plume near its source and after the long-range transport to Lindenberg.

To provide some background, measurements of $SO_2$ amounts and volcanic plume heights by local ground-based instruments

and by the TROPOMI onboard the Sentinel-5P satellite have been compared over the entire eruptive period (September to December 2021) first. For $SO_2$ total vertical column, data obtained with the Brewer spectrophotometer of the Izaña Observatory on 144 km distance to the vent have been used. Good qualitative agreement throughout the entire eruptive period of Tajogaite volcano has been found. A quantitative comparison has proven difficult because of the different FOVs of both instruments and the associated uncertainties. For volcanic plume height, measurements performed with several lidar instruments stationed

across La Palma have been employed. When compared to the TROPOMI $SO_2$ layer height retrieval, differences have been found to be in the range of 1-3 km. In view of the methodological differences and adverse effects such as the temporal mismatch between the measurements, differing FOVs of the instruments, and interfering clouds, the results are satisfactory. The TROPOMI $SO_2$ layer height retrieval exhibits a slight low bias with respect to the lidar measurements, both datasets are correlated with a correlation coefficient of $r = 0.74$.

A new modeling approach based on TROPOMI $SO_2$ VCD measurements and the HYSPLIT trajectory and dispersion model had to be developed to establish the Tajogaite eruption as the source of the aerosol which was subsequently observed over Lindenberg after its long-range transport. This endeavor became necessary because the direct retrieval of the $SO_2$ layer height from the TROPOMI measurements was not possible due to the low $SO_2$ content of the air masses. The extensive model calculations have shown that the aerosol measured by RAMSES on 26-27 September 2021 could indeed be traced backed to

the Tajogaite eruptions around 23 September 2021. According to this analysis, the volcanic plume over Germany was at heights between 3 and 6 km (at the time of the TROPOMI measurement on 26 September) and was emitted at an injection height of about 2.5-5 km 3-4 days earlier. Both height ranges are in good agreement with the RAMSES measurement, and with the lidar and TROPOMI observations near the volcano, respectively.

The RAMSES measurements have been discussed in detail. The analysis of the particle elastic and inelastic optical properties

of the volcanic aerosol showed that it consisted of inorganic, small, solid and irregularly shaped particles, the presence of large aerosol particles such as volcanic ash or Saharan dust as well as wildfire aerosols could be excluded. A new retrieval approach has been developed to estimate the microphysical properties of the volcanic aerosol. The parameters of the assumed normalized monomodal and lognormal size distribution are retrieved from the measured particle lidar ratio and particle depolarization ratio, number density from the measured particle backscatter coefficient. For the first time, an optical particle model has been utilized

that assumes an irregular, non-spheroidal shape of the aerosol particles. According to the microphysical retrieval, the particles of volcanic origin likely had an isometric to slightly plate-like shape with an effective half of particle maximum dimension around 0.1 μm and a real part of the refractive index of about 1.51. Moreover, mass column values between 70 and 110 $\mathrm{mg\,m^{-2}}$, mean mass concentrations of 45-70 $\mathrm{\mu g\,m^{-3}}$, and mean mass conversion factors between 0.21 and 0.33 $\mathrm{g\,m^{-2}}$ at 355 nm have been retrieved. A detailed discussion suggests that possibly RAMSES observed, at least in part, volcanic secondary

sulfate aerosol which was produced by gas-phase homogeneous reactions during the transport of the air masses from La Palma to Lindenberg.

*Author contributions.* PH conceived the study with the help from DL, and wrote part of the manuscript. PH and FL carried out the TROPOMI and Lindenberg HYSPLIT analysis and retrievals. JR performed and analyzed the RAMSES measurements, developed and applied the microphysical retrieval scheme, and wrote part of the manuscript. BW performed the TROPOMI HYSPLIT analysis and wrote Sect. 4.2.

AR provided the IZO Brewer data and support. AB and OG provided the AEMET ash layer height dataset. All authors contributed to the interpretation of the results and writing of the paper.

*Competing interests.* The authors declare that they have no conflict of interest.

*Disclaimer.*

*Acknowledgements.* The authors would like to thank the PIs and technical staff of the lidar instruments on the Canary Islands for the datasets

of altitude of the dispersion plume, namely:

- **CommSensLab, Dept. of Signal Theory and Communications, Universitat Politècnica de Catalunya (UPC), Spain:** Michaël Sicard, Constantino Muñoz-Porcar, Adolfo Comerón, Alejandro Rodríguez-Gómez

- **Atmospheric Research and Instrumentation Branch, Instituto Nacional de Técnica Aeroespacial (INTA), Spain:** Carmen Córdoba-Jabonero, María Ángeles López-Cayuela, Clara Carvajal-Pérez

- **ONERA, The French Aerospace Lab, Universite de Toulouse, France:** Andrés Bedoya-Velázquez, Romain Ceolato

- **Group of Atmospheric Optics, Universidad de Valladolid, Spain:** Roberto Román

- **Vaisala Oyj, Finland:** Reijo Roininen

- **INFN-GSGC L'Aquila and CETEMPS-DSFC, Università degli Studi dell'Aquila, Italy:** Marco Iarlori, Vincenzo Rizi, Ermanno Pietropaolo

- **INFN Napoli, Complesso Universitario Monte Sant'Angelo, Italy:** Carla Aramo

The authors thank the European Brewer Network (http://rbcce.aemet.es/eubrewnet/) for providing access to the data.



*Financial support.* This study was partially funded by the DLR projects S5P and INPULS (KTR 2472046 and 2472922). AB and OG acknowledge the support of ACTRIS Research Infrastructure Project by the European Union's Horizon 2020 research and innovation programme through ACTRIS-IMP (grant agreement no. 871115) and ACTRIS Spain.

*Review statement.*



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
