# Peer review of "Analysis of the long-range transport of the volcanic plume from the 2021 Tajogaite/Cumbre Vieja eruption to Europe using TROPOMI and ground-based measurements"

_EGUsphere, 2024_

## Author Comment (AC1)

**Reply to RC1 – from 22 Jul 2024**

We would like to thank you very much for your constructive comment. It is indeed challenging to determine the SO2 layer height in the UV and only recently new technical developments haver enabled to do this operationally, nevertheless at the cost of accuracy. We are however trying to improve the retrieval and improve the accuracy.

Thank you for your comment about Figure 4. It was challenging to transport all the different information contents of the sub panels given the same spatial distribution of pixels. We agree that the chosen color palette is misleading. We have therefore changed the color palette of the lower left image to rainbow, and of the right image to yellow-red-black which is indeed much easier to distinguish.

**Reply to RC2 – Alessia Sannino from 18 Sept 2024**

We would like to thank you for taking the time to provide us with your constructive comments and suggestion, which we will implement in the revised manuscript. Please find our replies to your comments below:

Line 86: 2.600 km: 2.6 km or 2.600 m?

The TROPOMI swath on the ground is indeed 2.600 km. No corrections needed.

Line 87 (and elsewhere): I recommend a non-breaking space between units of measurement and numbers and also between symbols and numbers as in lines 178, 448/449 and elsewhere

Many thanks for your suggestion. We have replaced all spaces between units and numbers with non-breaking spaces.

Line 179: 50-m

We have corrected this in the updated manuscript.

Line 209: It would be interesting to quantify how dry the air was

We have corrected this in the updated manuscript.

Figures 8 and 9: the graphs are very dense with information and the presence of the scale only on the left graphs complicates the reading. I would suggest adding it also to the right graphs.

We have updated the graphs.

Line 359: please indicate which panel of the figure the variables refer to or alternatively indicate the abbreviated name of the variable in the graph title.

We have corrected the figure.

**Reply to RC3 –from 07 Oct 2024**

We would like to thank you very much for your comments and suggestions, which we will implement in the revised manuscript. Please find our replies to your review below:

Major comments:

1) The authors should stress a bit more in the Introduction what is the added value of this work, to the scientific community. Which scientific question will be answered here that was hazy for so long? In my opinion, the application of non-spheroidal and irregular shape modeling for retrieving the aerosol microphysical properties (even if applied in a monomodal distribution), is quite innovative.

Thank you for this recommendation. We have added the following text to the introduction: "For the first time, an optical particle model is utilized that assumes an irregular, non-spheroidal shape of the aerosol particles."

2) To my understanding it is difficult to discriminate volcanic ash from SO2 with a lidar instrument. The term "volcanic ash" should be change to "volcanic cloud" when referring to lidar aerosol classification.

"Actually, it is quite easy to distinguish between volcanic ash and SO2-containing particles (sulfates) with a lidar because the former exhibits high depolarization ratios (the gas SO2 cannot be detected with Raman lidars at all). Volcanic ash was observed close to the eruptions, so the term is appropriate when lidar measurements from the Canary Islands are discussed. By the time the plume reached Lindenberg, all ash was gone, and only fine-mode particles prevailed. Therefore, here the term 'aerosol' is used.

Minor comments:

1) line 17: I think "extinction-to-mass conversion factors" is more suitable here.

We compared our results to those of Cordoba-Jabonero et al (2023), and they used the term 'mass conversion factor'. For this reason we prefer not to change the wording

2) line 18: "volcanic secondary sulfate": Thus, the observations of volcanic ash from the lidar measurements stated above (line 10), is probably misleading. Please rephrase if needed. When the lidar-probed aerosol layer cannot be classified (i.e. primary or secondary sulfates or ash etc) consider using the term "volcanic cloud" instead of "volcanic ash".

See our answer to major comment #2

3) Section 2.2: Which products are used from each instrument (and with what uncertainty) ? Can you summarize them in table 1 ?

We have added the information which instrument operated at which station to table 1. We decided not to show the uncertainty in the table since otherwise we had to add it to each parameter retrieved, which would make the table unreadable.

4) Section 2.3, line 140: "extinction": Not clear if the aerosol extinction coefficient was obtained from elastic or inelastic scattering. Apart from inelastic extinction coefficient also elastic is used here? Please mention briefly here, the uncertainty of the products provided by RAMSES .

This is probably a misunderstanding. The (particle) extinction coefficient is an elastic particle property because it describes the attenuation of light at the laser wavelength. It is either retrieved by some variant of the Klett method using an elastic lidar signal or it is measured using an inelastic (vibrational-)rotational Raman signal. In the case of RAMSES, a pure-rotational Raman signal around

356 nm is employed. Information on the statistical measurement uncertainties have been added to the text.

6) line 146: "Only measurements are considered that were taken close to the volcanic vent." -> "Only measurements taken near the volcanic vent are considered."

We have corrected this in the updated manuscript.

7) line 159: "differing"-> "different"

We have corrected this in the updated manuscript.

8) Figure 2: There are some negative values of SO2 VCD (in DU) as taken by the Brewer. What is their physical meaning?

This is related to the high noise level of the SO2 retrieval. When there is low or no SO2 in the atmosphere, the noise can cause negative SO2 values.

9) Figure 3:

a) Please consider providing the points on the graphs with their uncertainty (or variation).

We have added the uncertainties in the LH retrievals to Figure 3 and also added a description in the text. Note that the aerosol LH is about 258m, which is based on a validation against robust estimation of the height of the volcanic plume from IGN, Spain based on a video-surveillance camera monitoring network. , This IGN methodology was considered the reference value during the volcanic eruption and their values were used in the crisis Committees and in the VAAC reports.

b) Seems like the number of dots in panel c) are bit more in number compared to the number of x in panel a). Please cross-check if this is the case.

Yes, this is indeed the case. The reason for this is that sometimes for a given measurement time either no ground-based ash LHs or S5P SO2 LHs are available. For this case of course no difference can be calculated in panel b) and this point does not appear in panel c). Panel a) however shows all available measurements.

c) Are the cloud fractions and height difference correlated ?

The cloud fraction and height difference are not directly correlated: The CF can have an impact on the LH retrieval e.g. by shielding the SO2 cloud when the meteorological cloud is above (or mixed) with the SO2 layer. If the meteorological cloud is below the SO2 cloud, the higher (surface) albedo could lead to an increased sensitivity. The most reliable SO2 LH value can however be retrieved when there is no meteorological cloud in the pixel. This is clearly visible for low CF pixels showing low differences to the ash LH.

We have added a clarification in the updated manuscript

d) If the TROPOMI SO2 layer height is taken within coarser spatial resolution, how would this affect the correlation with the volcanic cloud retrieval from the ground-based lidars?

Since the ground-based lidars have a very narrow FoV compared to the TROPOMI spatial resolution, the correlation would be worse since the volcanic SO2 cloud could not be resolved properly and only an effective layer height would be retrieved. If TROPOMI would have a higher spatial resolution the correlation would be become better since about the same area of the volcanic cloud could be sensed.

8) line 262: Yes, volcanic ash may play a role in this underestimation. Is this within the limits reported in the scientific literature?

The underestimation of the SO2 LH in the presence of volcanic ash is a well know problem due to the light shielding effect of the ash particles. This effect can even yield SO2 LH being more than 5km different from the true LH. We have added a comment on this in the updated manuscript.

But on top of that, here the authors are comparing SO2 retrieval from a passive sensor with aerosol (volcanic cloud) retrieval from active sensors. I wonder if this also plays a role.

The type of sensing (active vs passive) does not play a role since in both cases the retrievals are based on the measured absorption of light (either sunlight or laser). The only difference is the measurement wavelength and which atmospheric constituents (ash or SO2) are observed

9) line 273: "lower"-> "less" and "lower than"->"below"

We have corrected this in the updated manuscript.

10) line 281: "the day of the start"-> "the starting day"

We have corrected this in the updated manuscript.

11) line 319: Is this due to the atmospheric conditions (e.g. wind speed etc), aerodynamics (e.g. particle size, etc)? Please elaborate more.

We have added an explanation for this.

12) line 364: "It can therefore be assumed... were not of spherical shape". Is this the the outcome of the droplet solidification?

Possibly, but one cannot tell. If the air parcels were dry all along their transport path, there would have been no need for the aerosol particles to solidify.

13) line 365: "the low δpar values near 2%": But these values indicate rather spherical of regular shape particles right? Provide also the relevant reference at the end of this sentence.

This is a common misunderstanding. Depolarization ratio does not only depend on shape but also on size. Even a highly irregular particles will not depolarize if it is small enough. So the conclusion at the end of the sentence stems from general scattering theory.

14) line 412: "...Vieja lava has a low SiO2 content, it is therefore .... ". Please provide the relevant reference for this statement.

We have added two references for this statement